# Midbrain projection to the basolateral amygdala encodes anxiety-like but not depression-like behaviors

Carole Morel [1,2✉], Sarah E. Montgomery[1,2,3], Long Li [2,3], Romain Durand-de Cuttoli[2,3], Emily M. Teichman[1,2,3], Barbara Juarez [1,2,3,4,5], Nikos Tzavaras[3,6], Stacy M. Ku[1,2,3], Meghan E. Flanigan[2,3,7], Min Cai[1,2], Jessica J. Walsh[1,2,3,8,9], Scott J. Russo [2,3], Eric J. Nestler [1,2,3], Erin S. Calipari [2,3,10], Allyson K. Friedman [1,2,11] & Ming-Hu Han [1,2,3,12✉]

Anxiety disorders are complex diseases, and often co-occur with depression. It is as yet unclear if a common neural circuit controls anxiety-related behaviors in both anxiety-alone and comorbid conditions. Here, utilizing the chronic social defeat stress (CSDS) paradigm that induces singular or combined anxiety- and depressive-like phenotypes in mice, we show that a ventral tegmental area (VTA) dopamine circuit projecting to the basolateral amygdala (BLA) selectively controls anxiety- but not depression-like behaviors. Using circuit-dissecting ex vivo electrophysiology and in vivo fiber photometry approaches, we establish that expression of anxiety-like, but not depressive-like, phenotypes are negatively correlated with VTA → BLA dopamine neuron activity. Further, our optogenetic studies demonstrate a causal link between such neuronal activity and anxiety-like behaviors. Overall, these data establish a functional role for VTA → BLA dopamine neurons in bi-directionally controlling anxiety-related behaviors not only in anxiety-alone, but also in anxiety-depressive comorbid conditions in mice.

[1] Department of Pharmacological Sciences, Icahn School of Medicine at Mount Sinai, New York, NY, USA. [2] Friedman Brain Institute, Center for Affective Neuroscience, Icahn School of Medicine at Mount Sinai, New York, NY, USA. [3] Nash Family Department of Neuroscience, Icahn School of Medicine at Mount Sinai, New York, NY, USA. [4] Department of Psychiatry and Behavioral Sciences, University of Washington Medical Center, Seattle, WA, USA. [5] Department of Pharmacology, University of Washington Medical Center, Seattle, WA, USA. [6] Microscopy Core, Icahn School of Medicine at Mount Sinai, New York, NY, USA. [7] Bowles Center for Alcohol Studies, University of North Carolina School of Medicine, Chapel Hill, NC, USA. [8] Department of Pharmacology, University of North Carolina at Chapel Hill, Chapel Hill, NC 27599, USA. [9] Neuroscience Center, University of North Carolina at Chapel Hill, Chapel Hill, NC 27599, USA. [10] Department of Pharmacology, Vanderbilt Center for Addiction Research, Vanderbilt University, Nashville, TN, USA. [11] Department of Biological Science, Hunter College at the City University of New York, New York, NY, USA. [12] Department of Mental Health and Public Health, Faculty of Life and Health Sciences, Shenzhen Institute of Advanced Technology, Chinese Academy of Sciences, Shenzhen, Guangdong, China. ✉email: carole.morel@mssm.edu; ming-hu.han@mssm.edu

Anxiety disorders are the most common psychiatric illnesses, afflicting 273 million people worldwide[1–4]. The symptoms of anxiety disorders are heterogeneous and highly complex, and the causes are poorly understood. Additionally, a substantial number of patients suffering from anxiety disorders also present with depressive-like symptoms, which show >60% overlap[5–8] and are frequently associated with greater severity and complexity[5–8], impeding the investigation of the neural dysfunctions that underlie anxiety per se. However, few neurobiological investigations have delved into the neural circuit mechanisms underlying anxiety disorders in anxiety-alone versus anxiety-depression conditions.

The midbrain dopamine system is crucial for adaptive behaviors[9–11] and the maintenance of healthy brain functions[12–15]. An increasing body of evidence from human brain imaging and preclinical animal studies implicate the mesocorticolimbic dopaminergic system in anxiety disorders and major depression[11,12,16,17]. Previous investigations have established that VTA dopamine neuron sub-circuits to the nucleus accumbens (NAc) and medial prefrontal cortex (mPFC) differentially regulate depressive-like behaviors[13–15,18]. However, the role of VTA sub-circuit function in anxiety has not yet been defined in the context of anxiety-alone or co-occurrence with depression. In addition to NAc and PFC, VTA dopamine neurons also project to amygdala sub-nuclei, including the BLA[19–22], a crucial structure in emotion and anxiety processes[23–27]. While it is known that the dopamine system plays a powerful role in the modulation of BLA activity[28–30], the contribution of VTA → BLA dopamine neurons in the emergence and expression of anxiety-like behaviors in the context of chronic stress exposure remains elusive.

The chronic social defeat stress (CSDS) paradigm induces heterogeneous individual behavioral phenotypes amongst socially stressed mice[31,32]. In this paradigm, while all mice are exposed to equivalent social stress, only a subset of them are susceptible to depression-like endpoints and develop social avoidance, anhedonia, disrupted circadian rhythms, and deficient reward learning[16,31–34]. The remaining defeated mice, termed resilient, do not develop these endpoints. Interestingly, both susceptible and resilient mice to depression-related outcomes develop anxiety-like phenotypes, suggesting that there are divergent neural circuits that underlie these behaviors[16,31–33]. While the brain mechanisms underlying susceptibility or resilience to depressive-like behavioral abnormalities is an active field of research, little is known about the neural mechanisms underlying the anxiety-like phenotype under the conditions of anxiety-alone and comorbidity with depression. To investigate anxiety-related mechanisms by using the CSDS paradigm, here for convenience we relabeled the depression-susceptible subgroup as "AD mice" that exhibit Anxiety- and Depressive-like phenotypes, and the depression-resilient subgroup as "A mice" that display solely an Anxiety-like phenotype. Comparing the neuronal dysfunctions of AD and A mice with stress-naïve control mice allowed us to parse the neurobiological substrates contributing to the singular and comorbid anxiety phenotypes from those regulating the depressive phenotypes.

Despite the need for more effective treatments with minimal side effects, the neural mechanism of anxiety—especially as it relates to comorbidity with depression—is not well understood. We took advantage of the CSDS-induced heterogeneous phenotypes, i.e., A mice and AD mice, when compared to stress-naïve control mice, to establish that a VTA → BLA dopamine circuit selectively controls anxiety—but not depression-like behaviors. First, we observed that CSDS induces anxiety-like behaviors independently from the induced depressive-like behaviors—i.e., social avoidance behaviors and reduced drive towards natural rewards, suggesting distinct brain circuit dysfunctions. Then, using circuit-dissecting ex vivo electrophysiological approaches, we show that anxiety- but not depressive-like behaviors are associated with hypoactivity of VTA → BLA dopamine neurons. Further, using circuit-specific in vivo fiber photometry, we show that the expression of anxiety-like behaviors is negatively correlated with VTA → BLA neuronal activity following CSDS. Finally, we demonstrate a causal link between such neuronal activity and anxiety-like—but not depressive-like—behaviors by employing optogenetic manipulation of the VTA → BLA circuit activity. Taken together, this study opens avenues of research on VTA → BLA circuit dysfunctions for the development of therapeutic treatments for both anxiety alone and anxiety comorbidity with depression.

## Results

**Anxiety behavior is independent of depressive-like behavior.** Following CSDS in C57BL/6J male mice, we first performed a social interaction (SI) test, which correlates strongly with a range of depressive-like phenotypes[31,32], and then performed an elevated plus maze (EPM) or open field test (OFT) to assess an anxiety-like phenotype (Fig. 1a, b). Compared to stress-naïve control (labeled as CTL in figures) and A mice, AD mice displayed social avoidance behaviors as reflected by decreased time spent in the SI zone when a social target was present and a lower (<100) SI ratio (Fig. 1c, d). On the other hand, both AD and A mice exhibited anxiety-like behaviors, i.e., decreased time and reduced number of entries in the EPM open arms (Fig. 1e–g and Supplementary Fig. 1a, b). Correlation analysis shows no significant relationship between SI behavior and time spent in the EPM open arms following CSDS (Fig. 1h), suggesting a lack of additive effect of depressive-like behaviors with anxiety-like behaviors. Also, we did not observe a significant relationship between SI behavior and time spent in the EPM open arms in stress-naïve CTL mice (Supplementary Fig. 1c).

To further assess anxiety-like behaviors, we performed an OFT in a separate cohort of mice after the CSDS and SI test (Supplementary Fig. 1d, e). We observed that both AD and A mice displayed reduced time spent in the open field center when compared to stress-naïve CTL mice (Fig. 1i–k and Supplementary Fig. 1), which again does not correlate with social avoidance behavior (Fig. 1l and Supplementary Fig. 1g), confirming that CSDS induces anxiety-like behaviors in both AD and A mice independently of depressive-like behaviors. Further, we observed that the SI behavior prior to CSDS does not correlate with SI/avoidance behavior after CSDS (Supplementary Fig. 1h, i). Conversely, we observed that the basal level of anxiety-like measures before CSDS significantly correlates with anxiety-like measures after CSDS (Supplementary Fig. 1j), indicating that CSDS exacerbates baseline levels of anxiety. As expected, stress-naïve CTL mice displayed stable behaviors before and after the CSDS paradigm (Supplementary Fig. 1k, l).

To further measure depressive-like behaviors following the CSDS paradigm, we performed female urine sniffing test (FUST)[35] and a sucrose preference test (SP) in a separate cohort of mice allowing for the assessment of reward-seeking behaviors[35] and anhedonia prior to the EPM and OFT tests (Supplementary Fig. 2a). We observed that AD mice exhibited a reduced preference for female urine as well as for sucrose after CSDS when compared to CTL and A mice (Supplementary Fig. 2b). Our correlation analyses show that following CSDS, preferences for both female urine and sucrose correlate with SI behaviors, but not with the time spent in the EPM open arms (Supplementary Fig. 2c). Conversely, we observed that the reduced time spent in EPM open arms and in OFT center correlate with each other in both AD and A mice (Supplementary Fig. 2d). Similarly, the

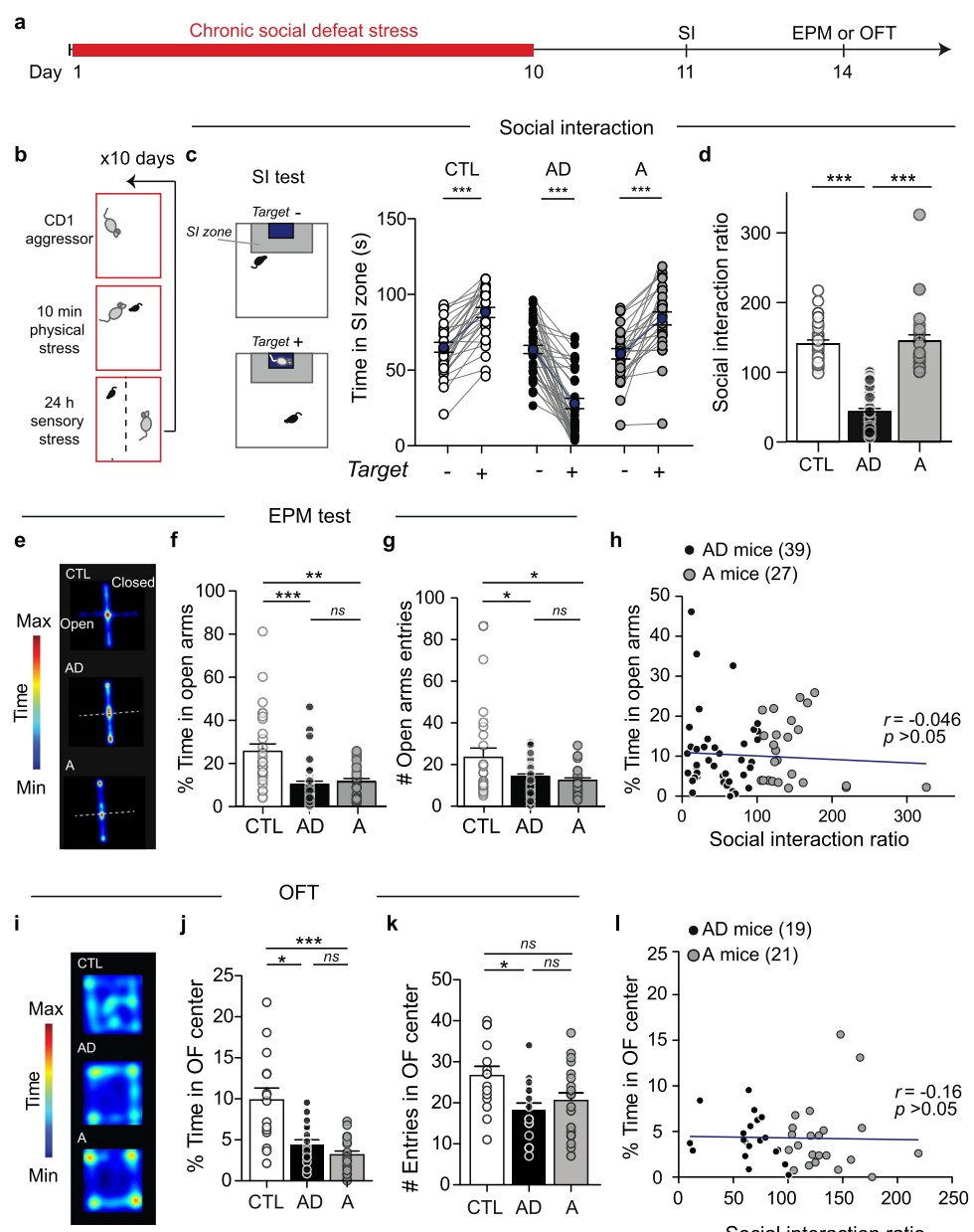

**Fig. 1 CSDS induces anxiety-related behaviors independently of social avoidance behavior. a** Experimental timeline for social interaction (SI) and elevated plus maze (EPM) or open field (OFT) tests. **b** Schematic of CSDS paradigm. **c** SI behavior in stress-naïve control (CTL) and socially defeated mice: AD mice display depressive-like social avoidance behavior and anxiety-like phenotype, and A mice exhibit only anxiety-like behavior (see data panels below), blue circles represent mean ± s.e.m. (Paired $t$-tests, $n = 28$ CTL $t = 9.02$ $p = 1.22e−09$, $n = 39$ AD $t = 10.38$ $p = 1.27e−12$, and $n = 27$ A mice $t = 6.78$ $p = 3.38e−07$, examined over 3 independent replicated experiments). **d** Resulting social interaction ratio, another way to analyze SI behavior that measures stable, relative time spent in SI zone (mean ± s.e.m., Kruskal–Wallis test, $H_{2/91} = 67.37$; $Z = 6.997$ $p < 0.001$; $Z = 6.80$ $p < 0.001$; $Z = 0.114$ $p = 0.99$). **e** Heatmap representation of the time spent in EPM compartments in CTL, AD, and A mice. **f** Time in EPM open arms (Kruskal–Wallis test, $H_{2/91} = 22.36$; $Z = 4.585$ $p < 0.001$; $Z = 3.432$ $p = 0.002$; $Z = 0.8381$ $p = 0.99$) and **g** EPM open arm entries in AD and A mice compared to control mice (ANOVA, $F_{(2, 91)} = 5.258$ $p = 0.007$; $t = 2.763$ $p = 0.01$, $t = 2.912$ $p = 0.09$; $t = 0.403$ $p = 0.69$; $n = 28$ CTL, $n = 39$ AD and $n = 27$ A mice, examined over 3 independent replicated experiments, bars represent mean ± s.e.m.). **h** Pearson correlation analyses of the time in open arms with the social interaction behaviors of socially stressed mice ($n = 66$, $p = 0.74$). **i** Heatmap representation of the time spent in the open field arena in CTL, AD, and A mice. **j** Time (%) in open field center (Kruskal–Wallis test, $H_{2/52} = 17.55$, $p = 0.0001$; $Z = 2.728$ $p = 0.02$; $Z = 4.168$ $p = 0.0001$; $Z = 1.474$ $p = 0.42$) and **k** number of open field center entries (ANOVA, $F_{2/52} = 4.831$ $p = 0.01$; $t = 3.032$ $p = 0.01$; $t = 2.293$ $p = 0.05$; $t = 0.859$ $p = 0.39$; $n = 15$ CTL, $n = 19$ AD and $n = 21$ A mice, bars represent mean ± s.e.m.). **l** Pearson correlation analyses of the time in open field center with the social interaction behaviors of socially stressed mice ($n = 40$, $p = 0.017$). In all panels, two-sided statistical analyses and post hoc tests were performed, *$p < 0.05$, **$p < 0.01$, ***$p < 0.001$, ns $p > 0.05$, for $n$ number of C57BL6/J mice. See also Supplementary Figs. 1 and 2.

preferences for female urine and sucrose in stress-naïve CTL mice correlate with SI behaviors (Supplementary Fig. 2e) but not the time spent in EPM open arms and OFT center, whereas the time in EPM open arms and OFT center correlates with each other (Supplementary Fig. 2f). These results provide further evidence that depressive-like behaviors following CSDS correlate with each other but are independent of the expression of anxiety-like behaviors. Together, these results suggest that depressive- and anxiety-like behaviors following CSDS may emerge from distinct neural mechanisms.

**VTA → BLA dopamine neuron hypoactivity is related to anxiety-like behaviors.** The midbrain dopamine system originating from the VTA is known to be a key component in reward processing and reinforcement, and also to play a pivotal role in adaptive behaviors in response to acute and chronic stress exposure[9,10,31,36,37]. Following chronic stress, the induced VTA → mPFC and VTA → NAc neuronal dysfunctions result in depressive-like behaviors[13–15,18]. In particular, CSDS induces VTA → NAc hyperactivity selectively in AD mice that leads to both social avoidance behaviors and decreased sucrose preference[13–15,18], while such behavioral outcomes are absent in A mice. In addition to their well-characterized projection to the NAc[38,39], VTA dopamine neurons project to amygdala nuclei, including the BLA[19–22]. We first performed circuit-probing approaches to identify if VTA → BLA and VTA → NAc neurons emerge from two distinct neuronal projections. We used a dual viral strategy to selectively label VTA → BLA and VTA → NAc neurons by injecting mice with 1) retrograde AAVrg-hsyn-eGFP in the BLA, and 2) AAVrg-hsyn-tdTomato in the NAc (Supplementary Fig. 3a). A similar strategy was performed in TH-BAC-Cre mice to selectively label VTA → BLA and VTA → NAc dopamine neurons by injecting 1) retrograde AAVrg-hsyn-DIO-eGFP in the BLA and 2) AAVrg-hsyn-DIO-tdTomato in the NAc (Supplementary Fig. 3a). We observed that only 2.7% of the 2100 labeled neurons were dual labeled for both pathways (Supplementary Fig. 3b). In line with the previous studies[19,20,40], these results support the existence of segregated sub-populations of VTA → BLA and VTA → NAc neurons.

To selectively explore the functional alterations of VTA dopamine neurons projecting to the BLA, we injected green luma retrobeads[13,14,19] into this region prior to CSDS (Fig. 2a–d and Supplementary Fig. 3c, d). This approach allowed us to label VTA neurons projecting to the BLA and selectively record their spontaneous firing activity. Following surgical recovery, CSDS, and behavioral assessments, we performed ex vivo cell-attached electrophysiological recordings from retrobead-labeled putative VTA → BLA dopamine neurons, identified using previously established electrophysiological criteria[14,41], in brain slices of stress-naïve CTL, AD, and A mice (Supplementary Fig. 3d–g top panels). We also recapitulated this experiment targeting VTA → BLA dopamine neurons in a cell- and circuit-specific manner[13] by injecting a retrograding Cre-dependent AAVrg-EF1a-DIO-eYFP in the BLA of Th-BAC-Cre mice (Supplementary Fig. 3d lower panels). We found similar activity between VTA → BLA retrobead-labeled putative dopamine neurons in C57BL/6 J mice and virally-labeled VTA → BLA dopamine neurons in TH-BAC-Cre mice (Supplementary Fig. 3e–g); thus these groups were collapsed (Fig. 2e, f). We observed significantly lower spontaneous firing rates in both AD and A mice when compared to stress-naïve CTL mice. We then assessed the relationship between VTA → BLA dopamine neuronal activity with social behavior (Fig. 2g) and time spent in EPM open arms (Fig. 2h). We observed that VTA → BLA dopamine neuronal activity does not correlate with SI/avoidance behaviors (Fig. 2g and Supplementary

Fig. 3h, i) but does correlate strongly with the time (%) in EPM open arms (Fig. 2h and Supplementary Fig. 3h). We further found that the correlation between VTA → BLA dopamine neuronal activity and time (%) in EPM open arms observed in CTL mice is disrupted by CSDS-induced hypoactivity (Supplementary Fig. 3i).

We then performed whole-cell patch-clamp recordings in CTL, AD, and A mice to further characterize the cellular alterations induced by CSDS. We observed lower excitability and a higher rheobase of VTA → BLA dopamine neurons in both AD and A mice when compared to CTL mice (Fig. 2i–k). We also observed that VTA → BLA dopamine neurons have lower hyperpolarization-activated cation current, i.e., $I_h$ current[14], and a smaller difference between the peak and steady-state voltages in response to hyperpolarizing current, i.e., sag amplitudes[19,42], in both AD and A mice when compared to CTL mice (Fig. 2l, m). These results are in line with the lower VTA → BLA dopamine neuronal activity observed in AD and A mice, suggesting that CSDS alters intrinsic VTA → BLA dopamine neuron properties in both groups. Together, these behavioral and cellular investigations support a possible contribution of VTA → BLA dopamine neurons in selectively mediating anxiety-like behaviors following CSDS.

**In vivo VTA → BLA activity correlates with anxiety-like behaviors.** Upon observing ex vivo pathological adaptions in the firing rate of VTA → BLA neurons in AD and A mice, we then considered whether this circuit-specific hypoactivity led to an increased anxiety level overall or if this dysfunctional activity was time-locked in response to a specific anxiogenic context. However, the specific temporal role of VTA → BLA neurons can only be assessed in awake, behaving animals. Therefore, to explore if temporally defined neuronal activity signatures map onto behavior, we investigated VTA → BLA neuronal activity during the expression of SI and anxiety-like behaviors using fiber photometry in freely behaving mice[43–45] (Fig. 3a and Supplementary Fig. 4a). We used a dual viral strategy to selectively target VTA → BLA neurons by injecting mice with (1) retrograde AAVrg-Cre-eYFP in the BLA and (2) AAV-DIO-GCaMP6 in the VTA (Fig. 3b). An optic fiber was then surgically implanted directly above the VTA allowing us to record VTA → BLA neuronal population activity using the GCaMP6 fluorescent calcium sensor (Fig. 3b, c). Surgical implantation of the optic fiber had no effect on social behavior and time spent in EPM open arms (Supplementary Fig. 4b, c). To time lock the VTA → BLA neuronal activity with individual behaviors, a video-tracking system was synchronized with the fiber photometry system. In stress-naïve mice, we first examined how the calcium-dependent signal—a proxy for VTA → BLA neuronal population activity—varied with occupancy of the SI zone during the SI test. For each mouse, we determined the mean activity in the SI zone, as $\Delta F/F$ z-scored area under the curve (AUC)[46], and further assessed the number of calcium-dependent events[44,47,48] over the full SI test trials (Supplementary Fig. 4d, e). We found that VTA → BLA neuronal activity was not associated with the time spent in the SI zone, with or without the social target-present (Supplementary Fig. 4d, e). We further compared GCaMP6 signal dynamics when the mice entered the SI zone with the social target-absent or present (Supplementary Fig. 4f–h). We did not observe significant VTA → BLA neuronal activation when the mice entered the SI zone when the social target was present, compared to when the social target was absent. These results are consistent with our ex vivo electrophysiological recordings showing that VTA → BLA neuronal activity is not associated with SI behavior.

In contrast, we observed that VTA → BLA neuronal activity was positively correlated with the time spent in the open arms in

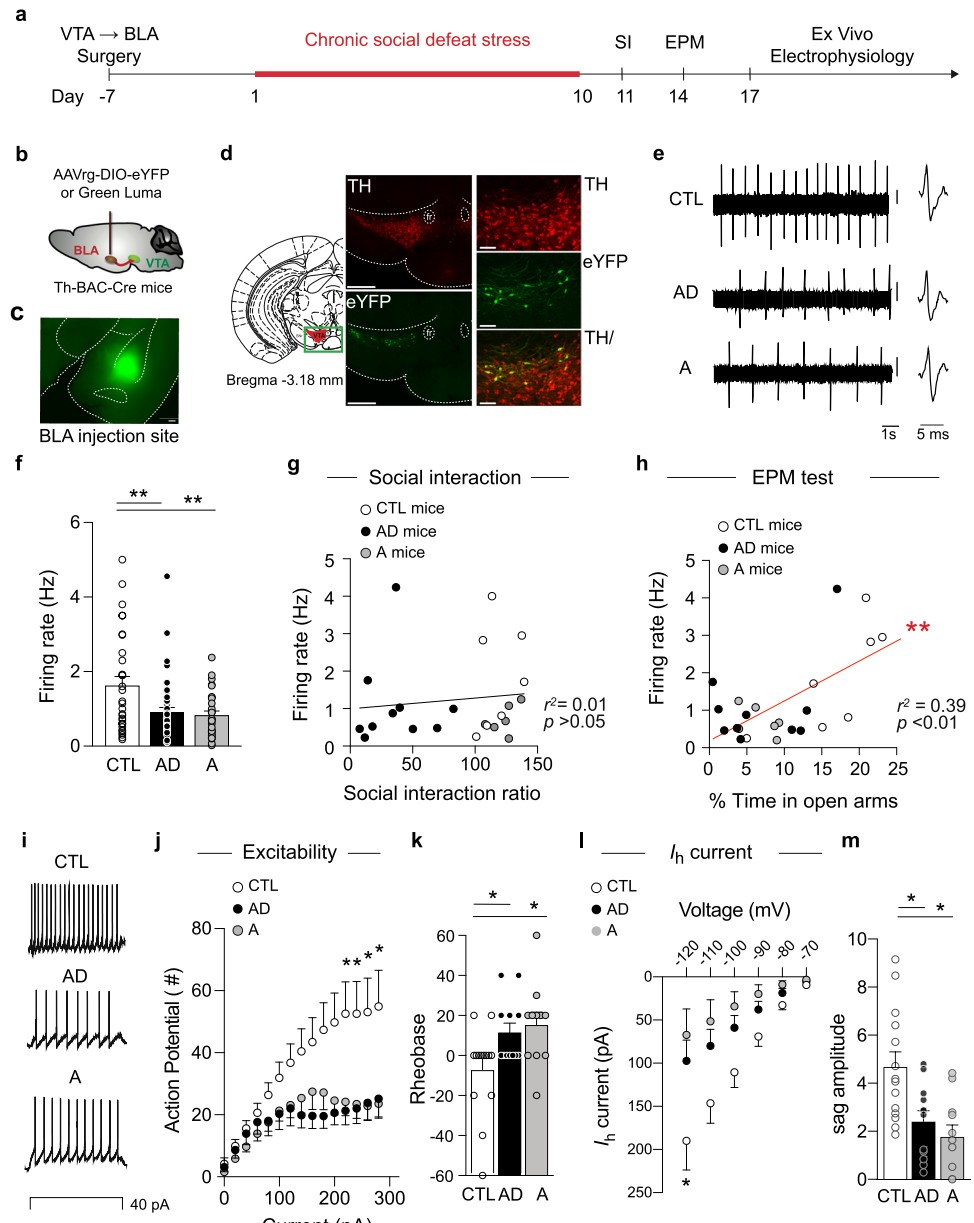

**Fig. 2 Anxiety-like behavior correlates with the hypoactivity of VTA → BLA dopamine neurons. a** Experimental timeline. **b** Schematic of the brain surgery to dissect VTA → BLA circuit. **c** BLA surgery injection site (scale bar=500 μm). **d** Morphological validation showing the targeted VTA → BLA dopamine neurons in TH-BAC-Cre mice injected with AAVrg-DIO-eYFP (scale bar = 500 and 100 μm, representative images of the 23 recorded mice). **e** Sample traces of ex vivo cell-attached recordings from CTL, AD, and A mice (scale bar = 0.2 mV). **f** Spontaneous firing activity of VTA → BLA dopamine neurons in AD and A mice compared to control mice (mean ± s.e.m., ANOVA, $F_{(2, 104)} = 6.750$ $p = 0.0018$; post hoc test, $t = 3.48$ $p = 0.002$; $t = 3.50$ $p = 0.003$, $n = 30, 31, 45$ neurons, $n = 23$ combined C57BL6/J and TH-BAC-Cre mice injected with AAVrg-DIO-eYFP and Green Luma, respectively). **g** Pearson correlation analyses of VTA → BLA dopamine neuron firing with the social interaction behavior after CSDS ($p = 0.59$, 3–7 neurons per mouse, $n = 23$ combined C57BL6/J and TH-BAC-Cre mice). **h** Pearson correlation analyses of VTA → BLA dopamine neuron firing activity with the time in EPM open arms ($p = 0.0015$, 3–7 neurons per mouse, $n = 23$ combined C57BL6/J and TH-BAC-Cre mice). **i** Sample traces of ex vivo whole-cell recordings from CTL, AD, and A mice at a 40 pA step current injection. **j** VTA → BLA dopamine neurons excitability in AD and A mice compared to CTL mice following incremental steps in currents injections (20–280 pA; mean ± s.e.m., RM two-way ANOVA: group effect: $F_{(2, 33)} = 3.818$ $p = 0.021$; Interaction $F_{(28, 434)} = 3.164$ $p = 1.08e{-}07$; post hoc tests: $t = 2.41$ $p = 0.04$; $t = 2.53$ $p = 0.04$; $t = 1.95$ $p = 0.04$; $t = 2.63$ $p = 0.04$; $t = 1.64$ $p = 0.04$; $t = 2.52$ $p = 0.04$; $t = 1.72$ $p = 0.04$; $t = 2.25$ $p = 0.04$; $n = 11, 12, 14$ neurons/4, 5, 6 TH-BAC-Cre mice). **k** VTA → BLA dopamine neurons rheobase in AD and A mice compared to CTL mice (mean ± s.e.m., ANOVA: Group effect: $F_{(2, 33)} = 4.016$ $p = 0.013$; post hoc tests $t = 2.43$ $p = 0.04$; $t = 2.85$ $p = 0.02$; $n = 11, 13, 14$ neurons/4, 5, 6 TH-BAC-Cre mice). **l** VTA → BLA dopamine neurons hyperpolarization-activated current, i.e., $I_h$ current in AD and A mice compared to CTL mice following incremental voltage steps (mean ± s.e.m., RM two-way ANOVA: group effect: $F_{(2, 33)} = 4.194$ $p = 0.017$; interaction $F_{(10, 175)} = 3.393$ $p = 9.7e{-}06$; post hoc tests $t = 2.22$ $p = 0.04$; $t = 2.71$ $p = 0.025$; $n = 11, 13, 14$ neurons/4, 5, 6 TH-BAC-Cre mice). **m** VTA → BLA dopamine neurons sag ratio in AD and A mice compared to CTL mice (mean ± s.e.m., ANOVA: group effect: $F_{(2, 32)} = 7.225$ $p = 0.001$; $t = 3.04$ $p = 0.009$; $t = 3.79$ $p = 0.002$, $n = 11, 13, 14$ neurons/4–6 TH-BAC-Cre mice). In all panels, two-sided statistical analyses post hoc corrected tests were performed, $*p < 0.05$, $**p < 0.01$. See also Supplementary Fig. 3.

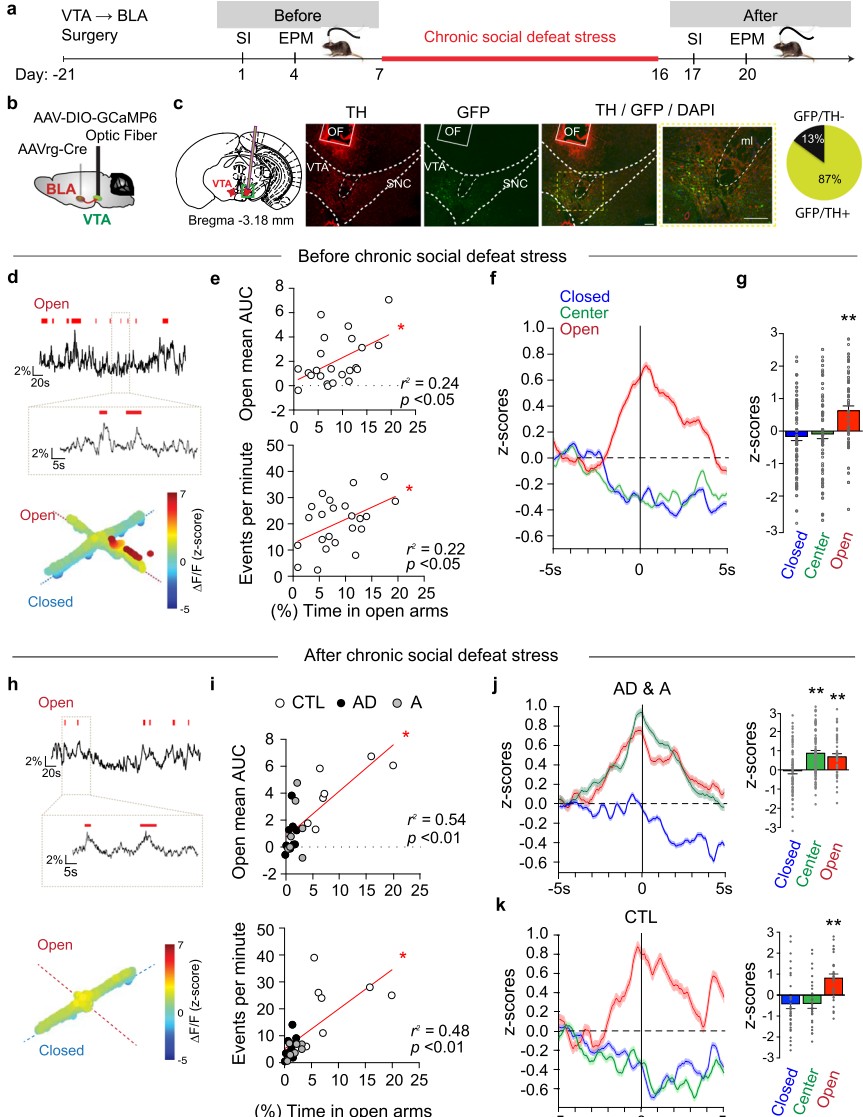

**Fig. 3 Anxiety-like behavior is associated with the dynamics of VTA → BLA neurons. a** Experimental timeline. **b** Schematic of the brain surgery targeting the VTA → BLA circuit. **c** Confocal image showing morphological validation of the placement of the optic fiber (OF) above the VTA and the co-expression of TH labeling with AAVrg-Cre-eYFP X AAV-DIO-GCaMP6 in the C57BL6/J mouse VTA (scale bar, 100 μm); quantification shows 87% colocalization (3–4 sections per mouse from 3 mice). **d–g** Before CSDS. **d** (Top) Sample traces of GCamp6 $\Delta F/F$ signal (bars represent the mouse in open arms). (Bottom) 3D representation of the GCamp6 $\Delta F/F$ upon the mouse position within the EPM. **e** Correlation analyses of the time spent in EPM open arms with (top, Pearson, $p = 0.02$) the mean open arms AUC VTA → BLA activity and (bottom, Pearson, $p = 0.02$) with the number of events per minute ($n = 23$ mice, examined across 3 independent replicated experiments). **f** Dynamics of GCamp6 signal (mean $z$-scores ± s.e.m.) 5 s before and 5 s after the mice enter EPM closed arms, center and open arms. **g** Averaged GCamp6 $z$-scores across stress-naïve mice within a 1-s bin ($-0.5$ to $+0.5$ s) time-locked to EPM closed arms, center or open arms entries (mean ± s.e.m., ANOVA, $F_{2/276} = 10.94$ $p = 2.6e-05$; $t = 4.32$ $p = 2.4e-05$; $t = 3.88$ $p = 1.7e-04$, $n = 80, 98, 100$ epochs). **h–k** After CSDS. **h** (Top) Sample traces of GCamp6 $\Delta F/F$ signal (bars represent the mouse in EPM open arms). (Bottom) 3D representation of the GCamp6 $\Delta F/F$ upon the mouse position within the EPM. **i** Correlation analyses of the time spent in EPM open arms with the mean open arms AUC VTA → BLA activity (top, Pearson, $p = 0.001$) and with the number of events per minute (bottom, Pearson, $n = 23$ mice, $p = 0.0002$). **j** (Left) Dynamics of GCamp6 signal (mean $z$-scores ± s.e.m.) 5 s before and 5 s after the socially stressed mice enter in EPM closed arms, center and open arms and (Right) related averaged GCamp6 $z$-scores across socially stressed mice within a 1-s bin ($-0.5$ to $+0.5$ s) time-locked to EPM closed arms, center or open arms entries in socially defeated mice (mean ± s.e.m., ANOVA, $F_{2/188}$ 11.79 1.5e−05; $t = 4.71$ $p = 0.001$; $t = 3.25$ $p = 0.003$, $n = 46, 65, 79$ epochs). **k** Same as **j** in stress-naïve CTL mice (mean ± s.e.m., ANOVA, $F_{2/92} = 10.59$ $p = 4.4e-05$, $t = 4.11$ $p = 0.001$; $t = 3.9$ $p = 0.001$; $n = 30, 30, 35$ epochs). In all panels, two-sided statistical analyses and corrected post hoc tests were performed, $*p < 0.05$, $**p < 0.01$. See also Supplementary Figs. 4 and 5.

stress-naïve mice: the higher the neuronal activity, the greater the time spent in the open arms over the full EPM trial (Fig. 3d, e and Supplementary Fig. 5a, b). We next examined the time-locked VTA → BLA GCaMP6 fluorescence dynamics upon entries in open arms, center, or closed arms. We observed that entries into the open arms were associated with increased VTA → BLA

neuronal activity, whereas entries to the closed arms or center compartments were not associated with changes in GCaMP6 fluorescence (Fig. 3f, g and Supplementary Fig. 5c–e).

We exposed the same cohort of mice to CSDS. In agreement with our behavioral and electrophysiological results, we observed that the decreased time spent in open arms following CSDS is

associated with a decrease of VTA → BLA neuronal activity (Fig. 3h, i and Supplementary Fig. 5f, g). These results are consistent with the hypothesis that VTA → BLA neuronal activity is a reflection of anxiety-like behaviors. Prior to CSDS, CTL, AD, and A mice maintained VTA → BLA activation upon entry into the EPM open arms (Fig. 3j, k and Supplementary Fig. 5h, i). Unexpectedly, we observed that entries to the EPM center were associated with VTA → BLA neuronal activation in AD and A mice that was absent in CTL mice (Fig. 3j, k and Supplementary Fig. 5h, i). Together, these results confirm that VTA → BLA neuronal activity in awake mice is altered in both AD and A mice and is significantly associated with the expression of anxiety-like behaviors.

**VTA → BLA neurons control anxiety-like behaviors.** Our electrophysiological and fiber photometry recordings identify an association between anxiety-like behavior and VTA → BLA neuronal activity. To test if there is a causal relationship, we performed bidirectional optogenetic manipulations by using inhibitory halorhodopsin (NpHR) or excitatory channelrhodopsin (ChR2)[13]. We first determined if the hypoactivity of VTA → BLA neurons is the causal mechanism that underlies the anxiety-related behaviors seen in both AD and A mice following CSDS. To selectively inhibit VTA → BLA neuronal activity with NpHR, we injected a retrograde AAVrg-Cre-eYFP vector into the BLA and a Cre-dependent AAV2-DIO-NpHR3.0-eYFP or AAV2-DIO-eYFP vector into the VTA (Fig. 4a and Supplementary Fig. 6a). We then performed functional validation in anesthetized mice with in vivo electrophysiological photo-tagged recordings of VTA → BLA putative dopamine neurons[13,49]. We found that a 582 nm light stimulation pattern at 0.1 Hz, 5 s pulse width decreased the spontaneous activity of VTA → BLA dopamine neurons by $45.8 \pm 5.2\%$ (Fig. 4b) to the same extent as seen in VTA → BLA dopamine neurons induced by CSDS.

In another group of mice, we performed the same viral injections and implanted an optic fiber above the VTA prior to the behavioral paradigm (Fig. 4a). We then exposed the mice to sub-threshold social defeat stress (Sub.D) to test if inhibiting the VTA → BLA circuit promotes the development of anxiety-like phenotypes. The Sub.D paradigm consisted of two brief episodes of social defeat stress within the same day after which the mice were returned to their home cage, priming the animal prior to optogenetic manipulations (Fig. 4a). A day later we performed the EPM test in Sub.D mice expressing NpHR or eYFP (Fig. 4c, d and Supplementary Fig. 6b, c) and in stress-naïve mice expressing NpHR (CTL-NpHR) or eYFP (CTL-eYFP; Supplementary Fig. 7a). We observed that NpHR-optogenetic inhibition decreased time spent in EPM open arms in Sub.D-NpHR mice as compared to Sub.D-eYFP mice (Fig. 4c and Supplementary Fig. 6b). We then prolonged the EPM test for 5 min while the laser was OFF and observed that without NpHR-optogenetic inhibition the Sub.D-NpHR mice behaved like the Sub.D-eYFP mice (Fig. 4d and Supplementary Fig. 6c). These findings are striking because they reveal the rapid induction of anxiety-like behavior that would normally require CSDS exposure. In contrast to the stress-primed mice, 0.1 Hz, 5 s pulse NpHR-optogenetic modulation pattern was insufficient to promote anxiety-like behavior in stress-naïve CTL-NpHR mice (Supplementary Fig. 7b, c); such anxiogenic effects were achieved when applying an NpHR-optogenetic manipulation with 1 Hz, 1 s pulse width during the entire test (Supplementary Fig. 7d, e).

We then tested the selective effect of VTA → BLA neuronal inhibition in anxiety-like but not depressive-like behaviors (Supplementary Fig. 8a). In a separate cohort of Sub.D mice, we observed that NpHR-optogenetic inhibition did not alter SI behavior and preference for female urine in Sub.D-NpHR mice when compared to Sub.D-eYFP mice (Supplementary Fig. 8b). In contrast, the NpHR-optogenetic modulation of VTA → BLA neurons decreased time spent in OFT center in Sub.D-NpHR mice when compared to Sub.D-eYFP mice (Supplementary Fig. 8b), confirming the anxiogenic effect of VTA → BLA neuronal inhibition. We extended the behavioral tests while the laser was off and did not observe significant behavioral differences between Sub.D-NpHR mice and Sub.D-eYFP mice (Supplementary Fig. 8c). Importantly, we did not observe prolonged behavioral effects after VTA → BLA neuronal inhibition as measured by sucrose preference and time spent in EPM open arms a week after the last optogenetic manipulation (Supplementary Fig. 8d).

To complete our bidirectional approach, we investigated if increasing VTA → BLA neuronal activity after CSDS prevented the anxiety-like phenotypes. We injected a retrograde AAVrg-Cre-eYFP vector into the BLA and a Cre-dependent AAV2-DIO-ChR2-eYFP or AAV2-DIO-eYFP vector into the VTA and implanted an optic fiber above the VTA prior to the CSDS paradigm (Fig. 4e and Supplementary Fig. 9a). We selected 5 pulses, 40 ms pulse width, delivered at 0.2 Hz pattern of ChR2-optogenetic stimulation (Fig. 4f) that we have previously shown elicits firing activity in VTA dopamine neurons[13,49]. Our in vivo electrophysiological photo-tagging recordings show that these stimulation parameters of one train every 5 s (each train constituted of 5 pulses at 20 Hz) increased the firing rate of VTA → BLA putative dopamine neurons by an average of $33\% \pm 9.17\%$ (Fig. 4f). First, following surgical recovery, CSDS, and SI test (Supplementary Fig. 9b), we performed an assessment of anxiety-like behaviors without VTA → BLA neuronal stimulation (Fig. 4g and Supplementary Fig. 9c-e) and found no basal behavioral differences between CSDS-ChR2 and CSDS-eYFP mice. Four days later we performed the EPM test on the same mice while applying the ChR2-optogenetic stimulation every 5 s over the 5 min trial. We observed an increased percentage of time spent in the open arm in CSDS-ChR2 mice when compared to CSDS-eYFP mice (Fig. 4h and Supplementary Fig. 9c, d, f). The anxiolytic effect of VTA → BLA neuronal stimulation was found in both AD and A mice (Supplementary Fig. 9c, d). These results confirm our hypothesis that increasing VTA → BLA neuronal activity after CSDS prevents the stress-induced increase in anxiety-like behavior.

We then tested the selective effect of VTA → BLA neuronal stimulation in anxiety-like but not depressive-like behaviors in a separate cohort of CSDS treated mice (Supplementary Fig. 10a). We observed that ChR2-optogenetic stimulation did not rescue the social avoidance behavior and reduced preference for female urine observed in AD mice (Supplementary Fig. 10b). When performing VTA → BLA neuronal stimulation in A-eYFP and A-ChR2 mice, we did not observe any effect on SI or preference for female urine (Supplementary Fig. 10c). In line with our physiological assessments, VTA → BLA neuronal stimulation increased the time spent in OFT center in both AD- and A-ChR2 mice when compared to AD- and A-eYFP mice (Supplementary Fig. 10b, c), confirming the anxiolytic effect of VTA → BLA neuronal stimulation. The effects of VTA → BLA neuronal stimulation on behavior were not sustained as we did not observe differences between CSDS-ChR2 and CSDS-eYFP mice in sucrose preference or time spent in EPM open arms a week after the last optogenetic manipulation (Supplementary Fig. 10d-e). Our optogenetic investigations unravel the selective role of VTA → BLA neurons in controlling anxiety-like, but not depressive-like, behaviors.

Given the temporal dynamics of VTA → BLA neuronal activity observed with fiber photometry in stress-naïve mice, we

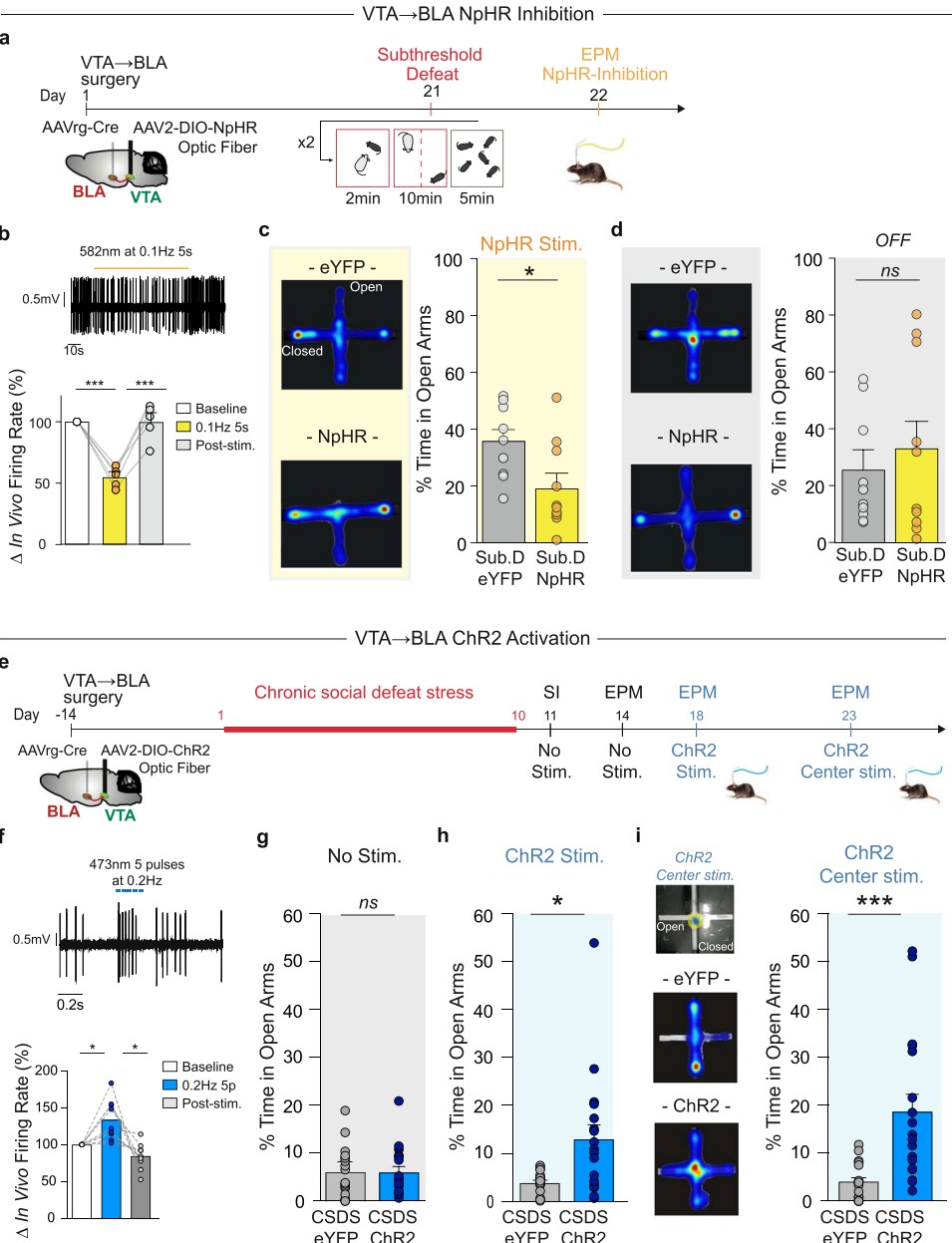

**Fig. 4 VTA → BLA neuronal activity controls anxiety-like behavior. a** Experimental timeline and schematic of subthreshold social defeat stress (Sub.D) experiment. **b** (Top) sample trace of in vivo photo-tagged recordings and (bottom) the mean ± s.e.m. averaged decreased firing activity of VTA → BLA dopamine neurons upon NpHR-optogenetic manipulation (0.1 Hz, 5 s pulse width, i.e., alternating 5 s ON and 5 s OFF during the 5 min EPM test period, ANOVA $F_{(2, 8)} = 48.44$ $p = 3.4e-05$; $t = 8.35$ $p = .001$; $t = 8.68$ $p = 0.001$). **c** Heatmap representation and time spent in EPM open arms of Sub.D experienced mice expressing eYFP (Sub.D-eYFP) or NpHR (Sub.D-NpHR) during the NpHR-optogenetic manipulation (t-tests, $t = 2.419$, $p = 0.027$, $n = 9,10$ C57BL6/J mice). **d** Heatmap representation and time spent in EPM open arms of the same mice while laser stimulation is not applied (t-test, $t = 0.554$, $p = 0.58$, $n = 9,10$ mice). **e** Experimental timeline and schematic of brain surgery to dissect the VTA → BLA circuit. **f** Sample trace of in vivo photo-tagging recordings and (bottom) the mean ± s.e.m. averaged increased firing activity of VTA → BLA dopamine neurons upon ChR2-optogenetic stimulation (RM ANOVA $F_{(2, 8)} = 12.11$ $p = 0.006$; $t = 3.67$ $p = 0.01$; $t = 3.54$ $p = 0.01$). **g** Time spent in open arms of mice exposed to CSDS expressing eYFP (CSDS-eYFP) or ChR2 (CSDS-ChR2) in VTA → BLA neurons while light stimulation is not applied (t-test, $t = 0.029$, $p = 0.97$, $n = 17,19$ C57BL6/J mice). **h** ChR2-optogenetic stimulation is applied during the 5 min EPM test (Mann–Whitney test, $U = 88$, $p = 0.01$, $n = 17,19$ mice). **i** ChR2-optogenetic stimulation is applied selectively to the EPM center. (Left) Schematic of the selective ChR2-optogenetic stimulation to the EPM center and heatmap representation of time spent in EPM compartments in CSDS-eYFP and CSDS-ChR2 mice. (Right) Resulting time spent in open arms of CSDS-ChR2 mice compared to the CSDS-eYFP mice (Mann–Whitney test, $U = 41$, $p = 0.001$, $n = 17,19$ mice). In all panels, data are represented as mean ± s.e.m.; two-sided statistical analyses and post hoc corrected tests were performed, *$p < 0.05$, **$p < 0.01$, ***$p < 0.001$. See also Supplementary Figs. 6–11.

hypothesized that VTA → BLA neuronal activation drives entry into the open arms of the EPM. We thus performed the EPM test while stimulating VTA → BLA neurons exclusively when the mice were located in the center of the EPM to test if this

optogenetic activation promotes open arm approaches, i.e., an anxiolytic response. We observed that the spatially constrained stimulation to the EPM center caused CSDS-ChR2 mice to increase the time spent in the open arms when compared to the

CSDS-eYFP mice (Fig. 4i and Supplementary Fig 9c, d, g). Importantly, the same anxiolytic effect of VTA → BLA neuronal activation was observed in both AD and A mice (Supplementary Fig. 9c, d). We also performed the EPM test coupled with ChR2-optogenetic stimulation in stress-naïve CTL-ChR2 and CTL-eYFP mice and observed that VTA → BLA neuronal activation did not alter the behaviors of CTL-ChR2 mice when compared to CTL-eYFP mice (Supplementary Fig. 11). Together, these results are important as they suggest that increasing the activity of VTA → BLA neurons induces an anxiolytic effect in both anxious-only and anxious-depressed subjects, thus offering avenues for therapeutic purposes.

## Discussion

A large body of evidence from preclinical studies and human brain-imaging investigations implicate midbrain dopamine neurons in adaptive behaviors versus behavioral abnormalities seen in psychiatric disorders[16,18,31,50] with an emphasis on how VTA dopamine neurons encode salient, rewarding, or aversive stimuli[10,13,14,51]. Recent studies have shown the contribution of VTA projections to the amygdala in encoding state-specific motivational salience[21], regulating approach/avoidance behavior toward threats[52], and have established the role of VTA projections in modulating basal amygdala (BA) activity during aversive conditioning[22]. In particular, activation of VTA → BA neurons was observed during tone-foot shock pairing[22]. Interestingly, inhibition of VTA → BA neurons during fear conditioning prevents the emergence of fear memories[22]. More recently the role of VTA projections to the amygdala in encoding the negative anxiogenic effect of acute nicotine has been established[53]. Here, by combining circuit-probing techniques with a unique behavioral model and functional measures, we demonstrate that VTA → BLA dopamine neuronal dysfunction is associated with anxiety-like behavioral abnormalities but not with depressive-like behaviors, after CSDS. Using real-time VTA → BLA neuron fiber photometry recordings, we observed that VTA → BLA neuronal activity reflects the basal level and stress-exacerbated level of anxiety that is associated with VTA → BLA neuronal hypoactivity. Our optogenetic manipulations establish a causal role for VTA → BLA dopamine neurons in the expression of anxiety-like behaviors that can be rescued by stimulating those cells.

As previously established, CSDS induces anxiety-like behavior, which in a subset of mice is concomitant with depression-like endpoints, i.e., social avoidance, anhedonia, disrupted circadian rhythms, disrupted reward learning[16,31,32,34]. Here, we observe that the expression of social avoidance behavior, a robust behavioral marker for a depressive-like phenotype[16,31,32,34], as well as disrupted preference for sucrose and female urine—both markers of hedonic and reward-seeking functions[35], are independent of the anxiety profile after CSDS. We also found that while vulnerability for a depressive-like phenotype could not be detected prior to CSDS, anxiety-like behaviors represent an exacerbation of a pre-existing anxiety trait. Together, these results support the hypothesis that there are divergent neural circuit alterations underlying depressive- versus anxiety-like behaviors.

Previous investigations have established that VTA → NAc dopamine neuron hyperactivity is causally linked to depressive-like behaviors observed in AD mice, while A mice exhibit homeostatic processes that maintain VTA → NAc circuit activity at levels similar to those in stress-naïve mice, thus preventing such behaviors[13,14]. Our current circuit-probing investigation shows that only 2.7% of VTA neurons are co-labeled for both VTA → NAc and VTA → BLA projections, supporting the idea that they are mainly segregated circuits with distinct functions in response to stress exposure and other stimuli. In line with these anatomical results, recent studies have identified distinct functions of VTA projections to the NAc or the amygdala in encoding the rewarding versus anxiogenic properties of acute nicotine injection, and activation of VTA projections to the amygdala prevents nicotine-induced anxiety-like behaviors[53]. Here, we demonstrate that VTA → BLA dopaminergic hypoactivity selectively controls the shared anxiety-like behaviors between AD and A mice but not the depressive-like behaviors unique to AD mice after chronic stress exposure. In sum, while the VTA → NAc circuit controls depressive-like behaviors[13], VTA → BLA dopamine neurons selectively regulate anxiety-like behaviors induced by CSDS. We further established that CSDS-induced VTA → BLA hypoactivity leading to anxiety-like behaviors does not require contextual association and pre-exposure to stressful events to induce the expression of anxiety-like behavior. Our results suggest that the physiological alterations of VTA → BLA neurons represent a general mechanism for the shared anxiety-like behaviors between AD and A mice, and further within psychiatric comorbidities. These results further demonstrate the importance of the midbrain dopaminergic system in regulating healthy brain function as well as the highly complex influence of VTA sub-circuits that differentially—and even oppositely—control anxiety-like and depressive-like behavioral abnormalities induced by chronic social stress.

We further define that VTA → BLA dopamine neuron hypoactivity following CSDS is associated with reduced excitability when compared to stress-naïve control mice suggesting that there are alterations in the intrinsic properties of VTA → BLA dopamine neurons. VTA dopamine neurons are heterogeneous in their target-specific projections, intrinsic characteristics, and responses to opioid, alcohol, and nicotine modulation[19,20,53–55]. $I_h$ currents contribute heavily to the intrinsic properties of VTA dopamine neurons[14,19,42]. In particular, VTA → NAc and VTA → BLA dopamine neurons have different $I_h$ currents. We previously established that altered VTA → NAc neuron $I_h$ currents are associated with depressive-like behaviors observed in AD mice[14]. Here, in line with our firing rate recordings and excitability measurements, we defined that VTA → BLA neurons in both AD and A mice have lower $I_h$ currents when compared to stress-naïve control mice. Together, these results provide a useful characterization of the VTA → BLA neuronal population as well as insight into how the hyperpolarization-activated cyclic nucleotide-gated (HCN) channels, which drive $I_h$ currents, contribute to the cellular mechanism resulting in VTA → BLA neuron hypoactivity in both AD and A mice. Further investigations to identify the precise basis for the reduced activity of VTA → BLA dopamine neurons following CSDS would remain critical for the development of anxiolytics.

More than two-thirds of patients suffering from anxiety disorders in their lifetime report a history of other mental disorders[5–8]. The co-occurrence of anxiety disorders with other syndromes—especially depression—challenges diagnostic and treatment strategies and questions the single disease framework of investigating treatment therapeutics. Here we identify VTA → BLA dopamine neuron hypoactivity as a biomarker and treatment target for anxiety-like behavior across single symptom profiles, i.e., A mice, and amongst more complex symptomatology such as co-occurring anxiety- and depressive-like behavior, i.e., AD mice. Our study thus provides insight for future clinical studies exploiting dopamine neuron circuits as a target for anxiety treatment, particularly within the context of anxiety and depression comorbidity.

## Methods

**Mice**. In this study, 7- to 9-week-old male mice were used. Heterozygous TH-BAC-Cre (GENSAT)[56] mice with a C57BL/6J genetic background were bred at the Icahn

School of Medicine at Mount Sinai. C57BL/6J mice were purchased from Jackson Laboratory and CD1 retired breeder mice were purchased from Charles River and were acclimated to the housing facility for 1 week prior to experiments. All mice were group-housed and maintained on a 12-h light/dark cycle under stable temperature (22–25 °C) and consistent humidity (50 ± 5%) with *ad libitum* access to food and water. Following CSDS, mice were then singly housed and maintained on a 12-h light/dark cycle with *ad libitum* access to food and water. All behavioral experiments were performed toward the end of the animal's light cycle. All experiments performed are approved by the Institutional Animal Care and Use Committee and comply with institutional guidelines for the Animal Care and Use Committee set forth by Icahn School of Medicine at Mount Sinai.

**Viral and tracing strategies.** For the selective expression of green retrobead fluorophores in the VTA → BLA projecting neurons, green retrobeads from Lumafluor Inc. were bilaterally injected (0.6 μl) into the BLA at a flow rate of 0.1 μl/min of C57BL/6J mice prior to the behavioral paradigms. For the cell- and circuit-specific expression of eGFP in the VTA → BLA dopamine-projecting neurons AAVrg-Ef1a-DIO-EYFP was bilaterally injected (0.6 μl at a flow rate of 0.1 μl/min) into the BLA of TH-BAC-Cre mice prior to the behavioral paradigms. For the circuit-probing approach investigating the potent co-projection of VTA neurons to both NAc and BLA, (1) TH-BAC-Cre mice were bilaterally injected (0.6 μl at a flow rate of 0.1 μl/min) into the BLA with AAVrg-hSyn-DIO-EGFP (addgene #50457) and into the NAc with AAVrg-FLEX-tdTomato (addgene #28306); (2) C57BL/6J mice were bilaterally injected (0.6 μl at a flow rate of 0.1 μl/min) into the BLA with AAVrg-hSyn-EGFP (addgene #50465) and into the NAc with AAVrg-CAG-tdTomato (addgene #59462). For the expression of GCaMP6 calcium sensors, NpHR and ChR2 light-activated pumps/channels or control fluorophores in VTA → BLA projecting neurons we used a dual viral system using (1) retrograding AAVrg.CMV.HI.eGFP-Cre.WPRE.SV40 (addgene #105545) bilaterally injected (0.6 μl at a flow rate of 0.1 μl/min) into the BLA and (2) AAVdj-EF1a-DIO-GCaMP6f, AAV9.CAG.Flex.GCaMP6s.WPRE.SV40, AAV2-EF1a-DIO-hChR2(H134R)-eYFP, AAV2-EF1a-DIO-NpHR3.0-eYFP, or AAV2-EF1a-DIO-eYFP obtained from Stanford University, Addgene, or the Gene Therapy Center Vector Core at the University of North Carolina for injection into the VTA. All viruses were freshly diluted in fresh and filtered 0.9% NaCl solution to obtain a final GC titer of $2–7 \times 10^{12}$.

**Stereotaxic surgeries and optic fiber implantation.** Mice were anesthetized with Xylazine/Ketamine (10 mg/kg; 100 mg/kg) and placed on a stereotaxic frame (Kopf Instruments). Ophthalmic ointment was applied to prevent eyes from drying and body temperature was maintained using regulated heated lamps. For a bilateral virus or retrobead injections under aseptic conditions, bregma was exposed, the head was flat-tested, and coordinates for VTA (7° angle, A/P −3.3 mm, D/V −4.6 mm; L/M ± 1.05 mm from bregma) and BLA (0° angle, A/P −1.6 mm, D/V −4.5 mm; L/M ± 3.1 mm from bregma) were calculated. Following injection (0.1 μl per minute) the needles were progressively removed. For the circuit-probing approach investigating the potent co-projection of VTA neuron to both NAc and BLA, two viral vectors were bilaterally injected in the NAc (rgTd-Tomato vectors: AAVrg-FLEX-tdTomato, addgene #28306 or AAVrg-CAG-tdTomato, addgene #59462; 10° angle, A/P + 1.4 mm, D/V -4.4 mm; L/M ± 0.5 mm from bregma) and in the BLA (rgGFP vectors: AAVrg-hSyn-DIO-EGFP, addgene #50457 or AAVrg-hSyn-EGFP, addgene #50465; 0° angle, A/P −1.6 mm, D/V −4.5 mm; L/M ± 3.1 mm from bregma). For the optic fiber implantation, the optic fibers were purchased from Doric for the fiber photometry (400/430NA) and optogenetic (200/240NA) approaches, implanted into the VTA (7° angle, A/P −3.3 mm, D/V −4.4 mm; L/M ± 1.05 mm from bregma) and secured with dental cement (C&B Metabond) and a screw to the skull (PlasticOne). Mice were then sutured and an antibiotic ointment was applied. Recovery from surgery was monitored for a minimum of 5 days post-surgery.

**CSDS paradigm.** Prior to the 10-day chronic social defeat paradigm, retired male breeder CD1 mice were screened for consistent aggressive and territorial behaviors. Screening for aggression required that a non-experimental C57BL/6J mouse was placed into the CD1 home cage for 1 min and the latency of the CD1 to attack the C57BL/6J mouse was recorded once per day for 3 days. CD1 mice with a daily attack latency of <1 min were used as experimental aggressors. Large mouse cages were modified to hold a plastic, perforated divider vertically through the cage with food and water on both sides of the divider. To perform the CSDS paradigm, experimental mice were introduced on the CD1 aggressor mouse side of the cage for 10 min. Following the social defeat with the resident CD1 aggressor mouse, experimental mice were physically separated for the rest of the day by the divider in the cage, allowing for continued sensory exposure. This procedure was repeated for 10 consecutive days—every day using an unfamiliar CD1 aggressor. The stress-naïve non-defeated control mice (CTL) were housed in pairs in a similar setting—1 CTL mouse on each side of a divided cage. Physical contact with another stress-naïve mouse was daily allowed 1 min per day, and then rotated to another cage as previously established[13–15,31,32].

**SI test.** We used an OFT box containing a wire mesh cage on one side. Testing conditions occurred under red-light conditions (<15 lux) in a room isolated from external sound sources. The open-field arena and wire-mesh enclosures were

thoroughly hand cleaned between mice with an odorless 5% ethanol cleaning solution. The socially defeated experimental mice or stress-naïve experimental control mice were individually placed in the OFT box and allowed to explore for 2.5 min (without social target present, known as the "No Target" phase). The experimental mouse was then removed, the open field was cleaned and an unfamiliar aggressor (CD-1) mouse was placed in the mesh cage (known as the "Target phase"). The experimental mouse was then returned to the open field and allowed to explore for another 2.5 min. Time spent interacting with the social target, locomotion and velocity were measured using a video tracking system (Ethovision). Experimental mice were then placed back into their home cage (singly housed). SI ratio was calculated as: SI ratio = ([time in SI zone "target"]/[time in SI zone "No target"])*100 as described previously[13–15,31,32]. The distance traveled and the velocity was analyzed during the "No target" phase to avoid potential biases due to the SI behaviors.

**EPM test (EPM).** The EPM was designed in black Plexiglass (L/W/D: 70/5/20 cm) and fitted with white surfaces to provide contrast. Testing conditions occurred under red-light conditions (<10 lux) in a room isolated from external sound sources. The EPM apparatus was thoroughly hand cleaned between mice with an odorless 5% ethanol cleaning solution. Mice were positioned in the center of the maze, and behavior was video tracked for a 5 min period[31]. Time in EPM compartments, locomotion, and velocity was measured using a video tracking system (Ethovision) set to localize the mouse center point at each time of the trial.

**Open field test (OFT).** Mice were placed in the open field arena (44 × 44 cm) for 5 min to compare the distance traveled and time spent in the peripheral zone compared to the center zone (10 × 10 cm). Testing conditions occurred under red-light conditions (<10 lux) in a room isolated from external sound sources. The EPM apparatus was thoroughly hand cleaned between mice with an odorless 5% ethanol cleaning solution. The mouse's activity—distance, velocity, and time spent in specific open field areas—was video-tracked and scored with the mouse Ethovision software[31].

**Sucrose preference (SP).** Mice were habituated to two bottles filled with drinking water (50-ml tubes with fitted ball-point sipper tubes) filled with drinking water for 1 day, and then mice were given access to a two-bottle choice of water vs 1% sucrose solution for the consecutive 2 days. Bottles were daily weighed and interchanged (left to right, right to left) to avoid biases from side preference. Sucrose preference was calculated as: SP = [sucrose solution consumed]/ [sucrose + water solutions consumed] × 100.

**Female urine sniffing test (FUST).** The FUST was adapted from the previous report[35] and performed to further assess reward-seeking behavior in stress-naïve CTL, AD, and A mice. Mice were initially placed in the 3-chamber arena (44 × 17 cm per chamber) for 3 min allowing habituation to the apparatus. Then mice were placed in the middle chamber while one surgical tapes squares were placed in the center of each side chamber: 1 absorbing 40 μl of female urine and 1 absorbing 40 μl of water. Mice were then allowed to freely explore each compartment of the 3-chamber apparatus for 5 min. The mouse activity—distance, velocity, and time spent in specific 3-chamber areas and center of each chamber (i.e., zone, 3 × 5 cm)—was video-tracked and scored with the mouse Ethovision software[31]. Testing occurred under red-light conditions (<10 lux) in a room isolated from external sound sources. The 3-chamber apparatus was thoroughly hand cleaned between mice with an odorless 5% ethanol cleaning solution. The side of the chamber containing the female urine was randomly attributed between mice. Preference for female urine over water sent was calculated as Female urine preference = [time spent in female urine zone]/[time spent in water zone].

**Group classification.** Following SI test and further FUST, SP, EPM, or OFT tests, CSDS-exposed mice were segregated into two subpopulations based on their social behavioral phenotypes: defeated mice that did not show a decrease in SI (interaction ratio ≥ 100) were termed resilient to depressive-like behaviors but expressing anxiety-like behaviors and were referred to as A mice. Mice that decreased their SI (interaction ratio < 100) were termed susceptible to depressive-like behaviors and expressing anxiety-like behaviors and here were referred to as AD mice. As we observed that the SI ratio correlates with other depressive-like behaviors—i.e., preference for female urine and sucrose (Supplementary Fig. 2), we used the SI ratio as an accurate measurement of depressive-like behavior across our study.

**Subthreshold social defeat stress (Sub.D).** To perform the subthreshold paradigm[13], the experimental mouse was placed directly into the home cage of a CD1 aggressor mouse for 2 min. During these 2 min, the mouse was physically attacked and chased by the CD1 mouse. After 2 min of physical interaction, mice were separated by a perforated Plexiglass partition, and the mice underwent 10 min of sensory stress. After 10 min of sensory stress, the experimental mouse was returned to its home cage for 5 min, before repeating physical stress and sensory stress with another aggressor. After two bouts of acute social defeats, the experimental mouse was returned to its home cage group-housed and underwent an

EPM test the next day coupled with optogenetic circuit manipulation. The related stress-naïve control mice for the Sub.D experiments were placed into the home cage of a non-experimental C57BL/6J mouse for 2 min. After 2 min of physical interaction, mice were separated by a perforated partition for 10 min. After 10 min of sensory contact, the mice were returned to their home cage for 5 min, before repeating physical and sensory contact with another C57BL/6J mouse. After two bouts of social contact, the experimental mice were returned to their home cage group-housed.

**Ex vivo electrophysiological recordings.** Acute coronal brain slices of VTA were prepared according to previously published protocols[13,14,57]. All recordings were carried out blind to the experimental condition. Male 8–12 week old mice were perfused with cold artificial cerebrospinal fluid (aCSF) containing (in mM): 128 NaCl, 3 KCl, 1.25 NaH$_2$PO$_4$, 10 D-glucose, 24 NaHCO$_3$, 2 CaCl$_2$, and 2 MgCl$_2$ (oxygenated with 95% O$_2$ and 5% CO$_2$, pH 7.35, 295–305 mOsm). Acute brain slices containing the VTA were cut using a vibratome microslicer (DTK-1000, Ted Pella) in sucrose-ACSF, which was derived by fully replacing NaCl with 254 mM sucrose, and saturated by 95% O$_2$ and 5% CO$_2$. Slices were maintained in the holding chamber for 1 h at 37 °C. Slices were transferred into a recording chamber fitted with a constant flow rate of aCSF equilibrated with 95% O$_2$ and 5% CO$_2$ (2.5 ml/min) and at 35 °C. Glass recording pipettes (2–4 MΩ) were filled with an internal solution containing (mM): 115 potassium gluconate, 20 KCl, 1.5 MgCl$_2$, 10 phosphocreatine, 10 HEPES, 2 magnesium ATP and 0.5 GTP (pH 7.2, 285 mOsm). Putative dopamine neurons were identified by their location and infrared differential interference contrast microscopy, and further electrophysiological criteria: regular and spontaneous action potentials with triphasic waveforms as previously described[14,31]. Recordings from VTA → BLA putative dopamine neurons were made from neurons labeled in green retrobeads in C57BL6J mice. VTA → BLA dopamine neuron recordings were made from eYFP virally tagged neurons in TH-BAC-Cre mice. The firing rate was recorded in the cell-attached mode. Similar to our previous studies[14,47], $I_h$ currents were recorded utilizing a whole-cell voltage-clamp protocol with a series of 3 s pulses with 10 mV command voltage steps from −120 to −60 mV with a holding potential at −60 mV[14,19,42]. To compare sag amplitudes of different DA neurons, the amplitudes of the current injections were adjusted in each cell to result in a peak hyperpolarization to ~80 mV, and the sag amplitude was determined as repolarization from ~80 mV to a steady-state value during the 1 s current injection[19,42]. Excitability measurements were recorded utilizing a whole-cell current-clamp protocol with a series of 1 s pulses with 20 pA command current steps from −100 to 280 pA with a holding current of 0 pA. Rheobases were determined by the minimal current step required to initiate an action potential. Series resistance was monitored during all recordings. Data acquisition and online analysis of firing rate and electrophysiological properties were collected using a Digidata 1440A digitizer and pClamp 10.2 (Axon Instruments)[13,14,57]. The sequence and timing of recordings were consistent throughout treatment groups. Neuronal firing rate activities were compared between the two retrograding methods for each treatment group using student $t$-test analyses. The average firing rate activities were not statistically different between the two retrograding methods and were combined in Fig. 2. To investigate the relationship between VTA → BLA dopamine neuronal activity with social behavior and time spent in EPM open arms, we calculated the average firing rate per mouse and then performed the correlations analyses between averaged firing activity and behavioral measurements.

**In vivo optrode photo-tagging recordings.** For in vivo optrode recordings using NpHR photo-modulation, C57BL/6J mice were injected bilaterally in the BLA (0° angle, A/P −1.6 mm, D/V −4.5 mm; L/M ± 3.1 mm from bregma) with 0.6 μl of rtgAAV.CMV.HI.eGFP-Cre.WPRE.SV40 and bilaterally 0.7 μl of Cre-dependent AAV2-EF1a-DIO-NpHR3.0-EYFP in the VTA (7° angle, A/P −3.3 mm, D/V −4.6 mm; L/M ± 1.05 mm from bregma). For in vivo optrode recordings using ChR2 photo-modulation, TH-BAC-Cre mice were injected bilaterally with 0.7 μl of Cre-dependent AAV2-EF1a-DIO-ChR2(H134R)-EYFP in the VTA at least 2 weeks prior to the optrode recordings to allow effective photo-modulation of VTA dopamine neurons. Single-unit extracellular recordings of VTA dopamine cells were performed in anesthetized (chloral hydrate 8%, 400 mg/kg i.p.) mice as described previously[49,58,59]. Glass electrodes (0.5% sodium acetate) coupled with optic fiber (200 μm, 0.2 N.A.) with an output of 5–10 mW, was lowered in the VTA according to stereotaxic coordinates (antero-posterior: −3 to −4 mm; medio-lateral: 0.1–0.7 mm; dorso-ventral: −4 to −4,8 mm from bregma). To distinguish dopamine from non-dopamine neurons, the following parameters were used: (1) firing rate (between 1 and 10 Hz); (2) action potential duration between the beginning and the negative trough superior to 1.1 ms[41,58–60]. Following a Shapiro test to determine the normality of the distributions, the spontaneous frequency was compared between groups using a Wilcoxon test, and the statistical significance was set at $P < 0.05$. Parameters of 0.1 Hz, 5 s pulse width, 5–10 mW using a 582 nm laser were sufficient to decrease the spontaneous activity of VTA dopamine neurons expressing the NpHR viral construct. Parameters of one train of stimulation, each train constituted by 5 pulses, 40 ms pulse width, 5–10 mW and separated by 10 ms light off[13,49] using a 473 nm laser, every 5 s (i.e., 0.2 Hz) were sufficient to elicit firing activity of VTA dopamine neurons expressing the ChR2 viral construct as previously established.

**In vivo fiber photometry recordings.** To analyze the bulk activity of VTA → BLA neurons via a GCaMP sensor in respect to the mice behaviors, the fiber photometry system was time-locked with the video-tracking system (Ethovision XT 11, Noldus) via transistor–transistor logic signals (TTLs). The fiber photometry system used two light-emitting diodes at 490 and 405 nm (Thorlabs), reflected off dichroic mirrors (FF495; Semrock) and coupled into a 400-mm 0.48 N.A. optical fiber (MFC_400/430-0.48; Doric). The light intensity at the fiber extremity ranged from 30 to 75 μW but was constant across trials and over days. The real-time fiber photometry signal was collected using a signal processor (Tucker–Davis Technologies) and acquired with open source OpenEx software 2.20 controlling an RX8 lock-in amplifier (Tucker-Davis Technologies). OpenEx (https://www.tdt.com/ support/downloads/; and https://www.tdt.com/component/openex-software-suite/), sinusoidally modulated each LED's output (490 nm at 211 Hz, and 405 nm isosbestic control at 531 Hz). The two output signals were then projected onto a photodetector (2151 femtowatt photoreceiver; Newport). The photoreceiver signal was sampled at 6.1 kHz, after which each of the two modulated signals was separated by the real-time processor for analysis[61]. Decimated signals were collected at a sampling frequency of 381 Hz[43,44,47] to perform the post-acquisition analyses.

Post-acquisition analyses were performed using custom programs and scripts in MATLAB based on generic codes from the Lerner and Gradinaru Labs that can be obtained from https://github.com/talialerner/ and https://github.com/GradinaruLab/. To compare VTA → BLA neuronal activity across animals and behavioral sessions, the 405 nm signal was used as a control channel to correct motion artifacts, auto-fluorescence, and bleaching. A least-squares linear fit was applied to the 405 nm control signal and fitted to the 490 nm signal. The fractional $\Delta F/F$ was determined over the full behavioral session as $\Delta F/F = ([490$ nm signal − 405 nm fitted]/405 nm Fitted) at each time point[43,44,47,62]. For the correlation analyses in Fig. 3, Supplementary Figs. 4 and 5, the resulting $\Delta F/F$ were z-scored over the entire behavioral session to allow for comparison between mice. Then AUC[46] was calculated using a trapezoidal method and computed for time spent in each apparatus compartment. To further characterize and compare VTA → BLA neuronal activity across animals and behavioral sessions (see Supplementary Figs. 4 and 5), we then identified calcium-dependent $\Delta F/F$ *peak* events during each behavioral session[44,47,48]. Events were determined significant when $\Delta F/F$ was superior to 2.91*median absolute deviation (the approximate 95% confidence interval MAD estimate for Gaussian data)[44,47,48]. The number of events was expressed as a number of events per minute and correlated with the mouse behaviors without assuming for Gaussian distributions (i.e., Spearman correlation).

To analyze time-locked neuronal activity in respect to the behavioral activity, the GCaMP signals and 405 nm signals were extracted from −5 s to +5 s around the onset of the relevant behavior (defined as $t = 0$ s): SI zone entries, transitions from EPM center to open arms, EPM center to closed arms and EPM closed arms to EPM center. Up to the five epochs per mouse were analyzed to avoid biases among mice and conditions (i.e., before and after CSDS). The resulting 10-s window signals were then z-scored and averaged allowing us to compare the GCaMP signal dynamics across behaviors and animals. Before averaging, each epoch was offset such that each z-score average from −5 s to −4 s equaled zero[45,63]. A 0.5-s sliding window was then applied to the slope of the averaged z-scores to observe the GCaMP dynamics across the different behaviors[64]. The summary z-scores quantification was calculated within a 1-s peri-event window centered to the behavior onset ($t = 0$ s) and compared between conditions. The same analyses were performed on the 405 nm control signal to ascertain that the observed activities were not due to any artifacts. In an attempt to ascertain that lack of GCaMP signal dynamic in respect to the social behavior was not due to the GCaMP kinetic characteristics, we used both GCaMP6s and GCaMP6f (see Supplementary Figs. 3 and 4). We observed that GCaMP6f displayed similar pattern dynamics to GCaMP6s across the behavior tests and we combined the data sets in Fig. 3 and Supplementary Figs. 3 and 4.

**Optogenetic approaches.** Mice used for VTA → BLA neuron optogenetic manipulations were randomly selected to receive either AAV2-EF1a-DIO-hChR2(H134R)-eYFP, AAV2-EF1a-DIO-NpHR3.0-eYFP, or AAV2-EF1a-DIO-eYFP viral injections before the behavioral paradigms. For the NpHR/582 nm opto-modulation experiment, 24 h following the sub-threshold social defeat stress, mice were connected to a dual optical fiber patch cord (200 μm, Doric) connected to a 582 nm yellow laser (Cristal laser and OEM laser)[13]. The mice were then placed in the center compartment of the EPM and a 0.1 Hz, 5 s pulse width, i.e., alternating laser ON for 5 s and then OFF for 5 s over the 5 min test period, 5–10 mW light stimulation pattern was applied using a pulse function generator (Agilent Technologies) for 5 min while the mouse behavior was being video-tracked (Ethovision). Then, the laser stimulation was stopped and the mouse behavior was video-tracked for another 5 min. Similar experiments were performed in stress-naïve mice (Supplementary Fig. 7d, e) while using a 1 Hz, 1 s pulse width, i.e., laser ON during the 5 min EPM trial. For the SI test, FUST and OFT coupled with NpHR opto-modulation mice were connected to the dual optical fiber and placed in the behavioral apparatus. Then the NpHR opto-modulation pattern (0.1 Hz, 5 s pulse, 5–10 mW light stimulation) was applied during the 150 s, 5 min, or 3 min tests of the respective SI test, FUST and OFT. Then the behavioral tests were prolonged, while laser OFF, by an additional 150 s, 5 min or 3 min tests of the respective SI test, FUST, and OFT. For the ChR2 473 nm opto-stimulation experiment, 4 days

following the assessment of the anxiety-like behaviors, mice were connected to a dual optical fiber patch cord (200 μm, Doric) connected to a 473 nm blue laser (OEM Laser System)[13,49]. The mice were then placed in the center compartment of the EPM and one train of stimulation was delivered every 5 s (i.e., 0.2 Hz; each train constituted by 5 pulses, 40 ms pulse width, 5–10 mW and separated by 10 ms light off) using a 473 nm light stimulation. After 4 days, the function generator was synchronized with the video-tracking system using TTLs. The mice were then placed in the center compartment of the EPM and the CHR2/473 nm light stimulation pattern was applied every 5 s exclusively when the mice were in the EPM center zone. The same pattern of opto-modulation and opto-stimulation were applied to the mice injected with the non-dynamic AAV2-EF1a-DIO-eYFP viral construct. For the SI test, FUST and OFT coupled with ChR2 optical stimulation mice were connected to the dual optical fiber and placed in the behavioral apparatus while laser off during the 150 s, 5 min, or 3 min tests of the respective SI test, FUST and OFT. Then the behavioral tests were prolonged, while laser ON and delivering one train of stimulation every 5 s, during an additional 150 s, 5 min, or 3 min tests of the respective SI test, FUST, and OFT.

**Immunohistochemistry**. Mice were perfused with 30 ml cold PBS and 30 ml 4% paraformaldehyde (PFA). The brains were collected and post-fixed in 4% PFA overnight, then treated with 30% sucrose at 4 °C for 2 days[57]. Brain tissue was sectioned with a thickness of 35–55 μm, rinsed with PBS, and then blocked with blocking buffer (3% BSA, 5% NDS, PBS with 0.2% Triton X-100) for 1 h. Sections were incubated with primary Anti-TH monoclonal antibody (Sigma-Aldrich 1:500, Cat# T2928, RRID:AB_477569) and anti-GFP (Invitrogen 1:2000, Molecular Probes Cat# A-6455, RRID:AB_221570) at 4 °C. The next day, sections were rinsed with PBS and then incubated with secondary antibodies (Alexa Fluor 488 1:1000, Jackson ImmunoResearch Labs Cat# 711-545-152, RRID:AB_2313584 and 647 1:500, Jackson ImmunoResearch Labs Cat# 715-605-150, RRID:AB_2340862) for 1 h, and then rinsed with PBS three times before mounting. Z-stack, tile scans, and single images were acquired using a Leica SP8 Confocal equipped with HyD and regular detectors (Leica Microsystems, Mannheim, Germany) using a 20×/0.75 N.A. HC PL APO CS2 objective. Luma-labeled and eYFP labeled with Alexa 488 antibody was excited using 488 nm argon-ion laser line and the fluorescence emission was collected from 491 to 600 nm. Alexa 647 was excited using a 633-nm HeNe laser line and the fluorescence emission was collected from 638 to 759 nm. Laser intensities and detector gain and offset (where necessary) values were adjusted to avoid detector saturation while maintaining a signal-to-noise ratio. The pinhole was set to 1 AU for 580 nm emission (56.7 μm), and pixel size was 1.517 × 1.517. Images presented are maximum intensity projection images from 16–20 z-slices (3 μm step). Images were analyzed using the Cell Profiler software (version 3.1.9). For the colocalization analyses, the slices were stained and mounted as previously described. Z-stack, tile scans, and single images were acquired using an EC Plan-Neofluar 10x/0.30 M27 objective on a Zeiss LSM 780. The pinhole was set to 1 AU and laser lines were used approximately as described above. Images presented and analyzed are maximum intensity projection images from 5 to 10 z-slices (~6 μm step size), with 3–6 samples per animal. Unbiased colocalization analyses within the VTA were performed using Cell Profiler software (version 3.1.9).

**Data analyses and statistics**. All behaviors were scored using the automated and unbiased Ethovision software. Experimenters analyzing the dataset were blind to the experimental conditions. All mice with an off-target viral injection or fiber implantation were removed from this study (n = 27). The statistic analyses were performed using R (version 3.3.3) and Graphpad (version 8, La Jolla, CA, USA) software. The statistic sample values were analyzed depending on the sample size, normality, and homoscedasticity of the sample. The normality of the distribution was assessed using Kolmogorov–Smirnov tests. The data fitting assumptions of the general linear model were subjected to linear regression, two-sided student t.tests, or multiple comparisons using a one-way, two-way, or repeated-measures ANOVA followed by post hoc two-sided $t$-tests with Dunnets and Bonferroni corrected for multiple comparisons $t$-test $p$ values. Analog nonparametric analyses were performed for datasets that did not follow a normal distribution or homoscedasticity using Spearman correlation analyses, or Kruskal–Wallis and Mann–Whitney two-sided statistical analyses. Statistical significance was set at 0.05.

**Reporting summary**. Further information on research design is available in the Nature Research Reporting Summary linked to this article.

## Data availability

All the data used in this study are included within the manuscript's figures or provided in the supplementary information section and Source Data files. Any additional data and information are available upon request to the corresponding authors, Drs. Carole Morel and Ming-Hu Han. Source data are provided with this paper.

## Code availability

Analyses were performed with MATLAB programs based on generic codes from the Lerner and the Gradinaru Labs that can be obtained from https://github.com/talialerner/ and https://github.com/GradinaruLab/. Any additional information is available upon request to the corresponding authors, Drs. Carole Morel and Ming-Hu Han.

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

## Acknowledgements

This work was supported by the National Institute of Mental Health Grant Nos. R56MH115409 and R01MH120637 to M.-H.H., and R01MH051399 to E.J.N., and R01MH120514 to C.M. and S.J.R., and by SC2 GM122646-03 to A.K.F., and K99DA054265 and T32DA007278-23 to B.J., and by the National Key R&D Program of China No. 2021ZD0202900 to M.-H.H. B.J. holds a Postdoctoral Enrichment Award from the Burroughs Wellcome Fund. This study was supported by the NARSAD Young Investigator Grant from the Brain & Behavior Research Foundation to C.M., and by the Hope for Depression Research Foundation to E.J.N.

## Author contributions
C.M. performed the behavioral assessments with the assistance of M.C. C.M. performed the ex vivo electrophysiology with the assistance of R.D.C., A.K.F., S.M.K., J.J.W. E.M.T. performed the whole-cell ex vivo electrophysiology. S.E.M., S.M.K., and B.J. assisted in surgeries. C.M. performed and analyzed the in vivo electrophysiology and optogenetic experiments. L.L. and S.E.M. assisted with the circuit-probing techniques. N.T. and S.E.M. performed the microscopy imaging and analyses. C.M. performed the fiber photometry experiments with the assistance of M.F. and technical supervision by E.S.C. who established the setup in S.J.R. and E.J.N. laboratories. C.M. and M.H.H. designed the experiments, interpreted results, and wrote the paper, which was edited by all authors.

## Competing interests
The authors declare no competing interests.
