## [Peer Review File · Nature Communications]

Reviewers' comments:

Reviewer #1 (Remarks to the Author):

This paper by Morel and colleagues combined behavioral, electrophysiological and fiber-photometric approaches to examine the role of an understudied VTA-BLA dopaminergic pathway in anxiety. Overall this is a nice contribution to the literature, but this study requires additional experiments to validate their conclusions, especially the one concerning the causal relationship between VTA-BLA neuronal activity and anxiety-like behavior.

The following is a list of notable weaknesses with the paper:

- Can the authors specify the selection criteria used to assign the entire population of mice exposed to CSDS to the AD and A group.
- In Figure 2, authors should clarify how a single firing rate value has been attributed to each mouse to perform correlative analysis.
- Authors should consider using *in vivo* electrophysiology (as they did in fig 4) or *ex vivo* patch clamp experiments to further characterize the electrophysiological parameters of VTA-BLA neurons. Those approaches will allow a better analysis of important parameters for VTA DA neurons (i.e., I_h current; SAG potential; F/I curve; A/N ratio; bursting activity), in control, AD and A mice.
- Authors should precise the type of retrograde tracer used in Figure 2C. All along the manuscript or figure legends, it is unclear if the mice used are th-BAC-cre mice or C57BL6/J mice.
- It is well known that VTA-DA neurons projecting to the nucleus accumbens extensively send extra-striatal axons (Beier et al; Cell 2015/Fallon J Neurosci 1981). Authors should quantify if VTA-BLA neurons project primarily to BLA, or if they have robust axon collaterals that target multiple regions. Since all manipulations of VTA-BLA DA neurons described in this study are performed at the level of the cell bodies, knowing about collaterals is of prime importance to correctly interpret the causal relationship between VTA-BLA neuronal activity and anxiety-like behavior after CSDS.
- Authors should provide a quantitative analysis of the impact of CHR2 activation on VTA DA neurons activity (as they did in figure 4b). In fact the protocol used by the authors is extremely strong (40 ms light ON followed by 10 ms light OFF, for 5 minutes), and it is important to control that the net effect of the manipulation leads to an activation of the VTA DA neurons.

Reviewer #2 (Remarks to the Author):

Morel et al. show that chronic social defeat stress (CSDS) induces anxiety-like behaviors distinct from depressive-like behaviors, and use *ex vivo* slice physiology to show that hypoactivity of VTA-BLA projection neurons is observed in mice with an anxiety-like phenotype (A) but not combined anxiety/depression-like phenotypes (AD). They also use fiber photometry to show that VTA-BLA DA activity is negatively correlated with anxiety-like behaviors, and optogenetics to show that activity in this circuit is causally related to anxiety-like behaviors.

The question that is posed is an important one — how mechanisms of psychopathology differ between individuals with anxiety vs. comorbid anxiety/depression, and the CSDS model seems like a good one to capture such differences given the heterogeneity in behavioral outcomes. However, while the experiments are carefully performed and outline a novel role of the VTA-BLA circuit in regulating anxiety-like behavior, I don't think the findings as presented rise to answering this ambitious question that is provided as the framework of the paper. This is largely because relatively little attention is paid to the depression-like phenotype, the majority of the effects observed seem to occur similarly in both A and AD mice, and optogenetic experiments appear to not differentiate between A and AD mice. Detailed comments and questions are provided below.

1. The authors use a single behavioral measure (social interaction) to determine a depression-like phenotype following CSDS. While they state that the SI phenotype correlates with other measures of depression-like behavior, they do not show results of any other depression-related assay. It is difficult to make a broad claim about depression using just the SI test. It would be nice to see the results of at least one other depression-related assay (e.g. for anhedonia), as they do for anxiety-like behavior using both the EPM and OFT.

2. The correlation plots in Figure 2g-h are not terribly convincing, yet they are used to support the claim that VTA-BLA dopamine neuron firing rate differentiates anxiety and depression phenotypes. It is a relatively small number of animals, and the trend appears to be driven largely by one or two control mice. Is there any significant correlation between firing rate and open arm time if you only plot just the stressed animals?

3. It is generally accepted that the EPM is not a very "repeatable" behavioral assay, meaning that upon second exposure to the maze animals tend to dramatically reduce open arm exploration. It is therefore concerning that the authors perform EPM twice on the same animals before and after CSDS. This could present a confound for the fiber photometry results — changes in circuit activity interpreted as due to stress may simply be due to adaptation to the same maze. The stress naive control is important for addressing this confound, but from the data presented it is difficult to tell which data points in Figure 3e correspond to those in Figure 3i (i.e. how the control animals' behavior and neural signals change upon second exposure to the EPM). Given the bizarre and surprising emergence of a large 'center'-related photometry signal after CSDS, this is cause for concern that the change in signal may be related to repeated exposure to the EPM. The authors should carefully control for this potential confound.

4. In Figure 3i, for the post-CSDS correlation plots, it is difficult to see the A and AD points since they are squished together on the x-axis. Could the authors provide additional plots of just the A and AD animals without controls so we can get a better sense of what these correlations look like in just the stressed animals?

5. In Line 228 the authors state: "the anxiogenic effect... is context dependent and requires a pre-exposure to stressful events to induce the expression of anxiety-like behavior". This is an intriguing result. Is it presumably because the stress is already inducing partial suppression of activity in these neurons? What would happen if you more completely inhibited VTA-BLA neurons in stress-naive animals (e.g. greater than ~45% observed with this pulsed laser inhibition)?

6. In Figure 4, does ChR2 stimulation differentially affect A and AD mice? Those groups are not differentiated in this figure.

7. Does sub-threshold social defeat stress not produce any behavioral effects on its own? In Figure 4c, it appears that the SubD eYFP group has similar if not even higher % time in open arms as the stress-naive controls in previous experiments.

8. Figure 4 - locomotion controls should be prominently shown for all optogenetic experiments. Optogenetic manipulation of VTA projections is known to produce locomotor effects when strong enough stimulation parameters are used, which could be a confound for changes in open arm time. These results are provided in the supplemental data, and it looks like there may be a trend toward increased locomotion with ChR2 stimulation. What are the exact p-values for these locomotor plots in Supplementary Figure 6f and 6g?

9. Do these VTA-BLA projection neurons send collateral projections to any other region?

Reviewer #3 (Remarks to the Author):

In this manuscript, the authors examine the intriguing question regarding separate circuits in CSDS mice that underlie depressive and anxiety phenotypes. They show that CSDS will induce depressive symptoms in 50% of the mice, but 100% show anxiety phenotypes. In this study, they use electrophysiology and fiber photometry/optogenetic methods to dissociate distinct dopamine pathways involved in these disorders, and show that the VTA DA system projecting to the amygdala is involved in the anxiety phenotype. This is a well-done study with potentially important results. Specific critique follows:

1. The use of a single metric for depressive behavior based on social interaction is likely not adequate, as there are a number of conditions that can impact social interaction, particularly given that the inducing stimulus is based on a negative social interaction.
2. It is not valid to say that there was a “significant decrease” in VTA DA firing, when one is not recording from the same neurons before and after a manipulation; it could just as readily been two different populations of neurons. A more parsimonious account would be to say that the average firing rate of the neurons recorded is lower after CSDS.
3. Figure 2H – the correlation between firing rate and time in open arms is driven entirely by the control mice; using all mice for a single correlation is not valid. This is also the case in Figure 3, where the effect is driven entirely by the control mice, and not by a comparison of A and AD mice.
4. The effects of optogenetic inhibition/activation of the VTA-BLA on behavior in the open arm maze is interesting. However, this may be confounded by the fact that DA in the BLA is involved in learning aversive behavior. Is the result truly an anxiety effect or a learning effect?

Reviewer #1 (Remarks to the Author):

This paper by Morel and colleagues combined behavioral, electrophysiological and fiber-photometric approaches to examine the role of an understudied VTA-BLA dopaminergic pathway in anxiety. Overall this is a nice contribution to the literature, but this study requires additional experiments to validate their conclusions, especially the one concerning the causal relationship between VTA-BLA neuronal activity and anxiety-like behavior.

We sincerely thank the reviewer for the very positive comments highlighting the importance of our study to the field and for the insightful recommendations. The reviewer's detailed comments and recommendations below helped us to refine our experimental design and expand the scope of our study. As such, we have addressed the weaknesses indicated by this reviewer below:

The following is a list of notable weaknesses with the paper:

- Can the authors specify the selection criteria used to assign the entire population of mice exposed to CSDS to the AD and A group.

We thank the reviewer's request to clarify the grouping criteria, which is very important in this study. The entire population of mice exposed to CSDS was assigned to the stress-exposed group (CSDS mice), while the stress-naïve mice were assigned to the control (CTL) group. CSDS mice have a reduced time spent in EPM open arms and in the OFT center when compared to CTL mice. These results confirm that CSDS induces anxiety-like behaviors when compared to stress-naïve CTL mice. The related statistical analyses comparing stress-exposed and stress-naïve mice are now provided in the revised Supplementary Figure 1a, d.

Then, the entire population of CSDS mice was further divided into the anxious & depressed group (labeled as AD mice) or anxious-only group (labeled as A mice), using the selection criteria is as follows: AD and A mice have social interaction ratio <100 and ≥ 100 , respectively, which has been explicitly defined in the revised Methods section (Line 583-589).

Finally, we observed a strong correlation between social interaction, sucrose preference and preference for female urine (Malkesman et al., 2011), the three metrics used to assess depressive-like behaviors (new Supplementary Figure 2c). These new results confirmed that social interaction ratio is a reliable metric to assess depressive-like behaviors induced by the CSDS paradigm.

- In Figure 2, authors should clarify how a single firing rate value has been attributed to each mouse to perform correlative analysis.

The single firing rate value attributed to each mouse was calculated as the average firing rate of multiple neurons (3-8 neurons) recorded from the same mouse. This important clarification was provided in the revised Methods section (Line 648-652).

Here for reference, we provided correlations between individual neuronal activity with the social behavior and the time spent in EPM open arms. While we consider that an averaged firing rate should be used as a single value per mouse, the data set provided in Figure A could be provided in Supplementary Information within the paper upon editor and reviewers' request.

Figure A. Pearson correlation analyses of individual VTA-BLA neuron firing rate with the social interaction ratio (Left) and the time spent in EPM open arms (Right).

- Authors should consider using *in vivo* electrophysiology (as they did in fig 4) or *ex vivo* patch clamp experiments to further characterize the electrophysiological parameters of VTA-BLA neurons. Those approaches will allow a better analysis of important parameters for VTA DA neurons (i.e, I_h current; SAG potential; F/I curve; A/N ratio; bursting activity), in control, AD and A mice.

Following the reviewer's insightful comment, we used our circuit-probing technique combined with patch-clamp whole-cell recordings to further characterize the electrophysiological properties of VTA-BLA neurons.

We defined that VTA-BLA neurons in both AD and A mice have a significantly reduced excitability when compared to control mice. This was further evidenced by a higher rheobase in the VTA-BLA neurons of both AD and A mice when compared to control mice. These new experimental results are now presented in Figure 2i-j.

Further, to define the intrinsic properties of VTA-BLA neurons, we assessed the hyperpolarization-activated cyclic nucleotide-gated (HCN) channel currents (i.e. I_h currents) and sag amplitudes as recommended by the reviewer. In line with our firing rate recordings, excitability measurements and rheobase data set, we observed that VTA-BLA neurons in AD and A mice have smaller I_h currents and sag amplitudes when compared to control mice. Together, these new neurophysiological results provide not only useful characterization of the VTA-BLA neuronal population but also identify HCN channels as a potential molecular contributor for the VTA-BLA neuronal hypoactivity in AD and A mice. These new experimental results are now presented in Figure 2k-m.

Finally, to better confirm the dynamics of VTA-BLA neurons *in vivo*, we opted for our circuit-specific fiber photometry approach in freely behaving mice as this technique allows us to time-lock VTA-BLA neuron activity with the mouse behaviors. We increased our sample size from $n=15$ to $n=23$ for a total of 3 cohorts. We confirmed that VTA-BLA

neuronal dynamic is associated with the time spent in EPM open arms but not with social interaction behaviors as shown in Figure 3 and Supplementary Figures 4 and 5.

- Authors should precise the type of retrograde tracer used in Figure 2C. All along the manuscript or figure legends, it is unclear if the mice used are th-BAC-cre mice or C57BL6/J mice.

We thank the reviewer for the pertinent comments. As requested, we have now provided the mouse genotype information within all figure legends of the revised manuscript.

- It is well known that VTA-DA neurons projecting to the nucleus accumbens extensively send extra-striatal axons (Beier et al; Cell 2015/Fallon J Neurosci 1981). Authors should quantify if VTA-BLA neurons project primarily to BLA, or if they have robust axon collaterals that target multiple regions. Since all manipulations of VTA-BLA DA neurons described in this study are performed at the level of the cell bodies, knowing about collaterals is of prime importance to correctly interpret the causal relationship between VTA-BLA neuronal activity and anxiety-like behavior after CSDS.

The reviewer raised an interesting question. Indeed, VTA dopamine neurons projecting to the NAc, the major target of the VTA projections, also send extra-striatal projections. To assess if VTA neurons projecting to the BLA also send striatal collaterals, we performed viral mediated-circuit tracing by injecting retrograding AAVs in the NAc allowing for somatic expression of tdTomato and retrograding AAVs in the BLA allowing for somatic expression of eGFP (Supplementary Figure 3a). Performing immunohistochemistry and confocal imaging analyses, we identified that only 2.7% of the 2100 labeled VTA neurons were labeled for both NAc and BLA pathways. These viral tracing results are in line with the previous reports (Ford et al., 2006) showing anatomical segregation between striatal VTA projections and VTA-BLA projecting neurons.

Additionally, our current results and previous publications (Chaudhury et al., *Nature* 2013; Friedman et al., *Science* 2014) show that VTA dopamine neuron somatic responses such as the firing rate and I_h currents to CSDS are different upon their projection targets. These distinct responses are summarized in Figure B below. In particular, VTA-BLA neurons are hypoactive in both AD and A mice, while VTA-NAc neurons are hyperactive only in AD mice when compared to control mice. Moreover, VTA-BLA neurons of stress-naïve mice display I_h currents, while VTA-PFC neurons do not. Together, in agreement with previous reports (Ford et al., 2006; Lammel et al., 2008; Margolis et al., 2008), our results support the idea that there is a low probability for VTA-BLA neurons to have robust collateral projections to the NAc and the PFC.

Figure B. Differential CSDS-induced alterations upon VTA dopamine neuron projections.

i. The hyper activity of VTA-NAc dopamine neurons exclusively observed in AD mice is associated with increased I_h currents. I_h currents are even more increased in A mice when compared to CTL mice. ii. VTA-PFC neurons are hypoactive exclusively in AD mice. (Chaudhury et al., *Nature* 2013; Friedman et al., *Science* 2014). iii. Our current study demonstrates that VTA-BLA neurons are hypoactive and have a reduced I_h currents in both AD and A mice following CSDS.

- Authors should provide a quantitative analysis of the impact of CHR2 activation on VTA DA neurons activity (as they did in figure 4b). In fact the protocol used by the authors is extremely strong (40 ms light ON followed by 10 ms light OFF, for 5 minutes), and it is important to control that the net effect of the manipulation leads to an activation of the VTA DA neurons.

We would like to thank the reviewer for pointing out the necessity for clarification and validation, as we understand that our previous terminology might have been misleading.

In our study, we used the optogenetic ChR2 stimulation pattern that we previously validated using *ex vivo* and *in vivo* electrophysiological recordings of VTA dopamine neurons (Chaudhury et al., *Nature* 2010; Juarez et al., *Nature Communications* 2017). The ChR2 stimulation pattern used consisted of one train of stimulation every 5 seconds (0.2 Hz); each train composed of 5 pulses with 40 ms pulse width, and separated by 10 ms light off. The last pulse (i.e. pulse #5) of the train was followed by 4750 ms light off before the following pulses train. Clarifications of the ChR2 optogenetic stimulation pattern were provided in the revised Methods section (Line 675-677 and Line 760-762), as well as in the revised Results section (Line 301 and Line 304, and Figure 4).

Importantly, as recommended by the reviewer, we performed additional *in vivo* photo-tagging recordings of VTA DA neurons allowing us to validate our approach and to quantify that this stimulation pattern resulted in a $33\% \pm 9.17$ increase of the VTA DA neuronal firing rate. These new results are now provided in the revised Figure 4f.

Reviewer #2 (Remarks to the Author):

Morel et al. show that chronic social defeat stress (CSDS) induces anxiety-like

behaviors distinct from depressive-like behaviors, and use ex vivo slice physiology to show that hypoactivity of VTA-BLA projection neurons is observed in mice with an anxiety-like phenotype (A) but not combined anxiety/depression-like phenotypes (AD). They also use fiber photometry to show that VTA-BLA DA activity is negatively correlated with anxiety-like behaviors, and optogenetics to show that activity in this circuit is causally related to anxiety-like behaviors.

The question that is posed is an important one — how mechanisms of psychopathology differ between individuals with anxiety vs. comorbid anxiety/depression, and the CSDS model seems like a good one to capture such differences given the heterogeneity in behavioral outcomes. However, while the experiments are carefully performed and outline a novel role of the VTA-BLA circuit in regulating anxiety-like behavior, I don't think the findings as presented rise to answering this ambitious question that is provided as the framework of the paper. This is largely because relatively little attention is paid to the depression-like phenotype, the majority of the effects observed seem to occur similarly in both A and AD mice, and optogenetic experiments appear to not differentiate between A and AD mice. Detailed comments and questions are provided below.

We thank the reviewer's very positive comments highlighting the necessity to investigate the neurobiological substrate underlying anxiety-like behavior using a model for anxiety/depression comorbidity. We are also grateful for the insightful comments and constructive recommendations, as we trust these comments and consecutive new experiments helped us to refine our experimental design and interpretation and importantly helped us to expand the scope and impact of our studies.

In particular, to address the reviewer's concern, we further characterized depressive-like behaviors following CSDS by adding behavioral assessments of anhedonia and reward-seeking behaviors. We also performed bidirectional optogenetic manipulations of the VTA-BLA neuronal activity, focusing on the impact of such manipulations onto depressive-like behaviors. These new results proposed the cellular and physiological alterations of VTA-BLA neurons as a general mechanism for the shared anxiety-like behaviors between AD and A mice, without significantly contributing to the depressive-like behaviors.

Below, we have addressed the comments and questions asked by the reviewer in details.

1. The authors use a single behavioral measure (social interaction) to determine a depression-like phenotype following CSDS. While they state that the SI phenotype correlates with other measures of depression-like behavior, they do not show results of any other depression-related assay. It is difficult to make a broad claim about depression using just the SI test. It would be nice to see the results of at least one other depression-related assay (e.g. for anhedonia), as they do for anxiety-like behavior using both the EPM and OFT.

We strongly agree with the reviewer suggestions and performed complementary analyses to further assess and characterize depressive-like phenotypes following CSDS in both AD and A mice.

In a new, separate cohort of mice presented in Supplementary Figure 2, we first used social interaction ratio to segregate AD mice and A mice. We then measured the preference for sucrose drinking solution over water as a measurement of anhedonia. In line with previous reports, we observed that AD mice (susceptible to depression), but not A mice (resilient to depressive outcomes), have a reduced sucrose preference when compared to stress-naïve control mice (Supplementary Figure 2b). Additionally, we performed a female urine-sniffing test (FUST) in the stress-naïve control, AD and A mice to measure reward-seeking behaviors, as defined by the ability of chemosensory cues related to sexual behavior, i.e. urine, to elicit an approach behavior (Malkesman et al., 2011). We observed that AD mice, when compared to stress-naïve control and A mice, have a reduced interest in pheromonal odors from the opposite sex.

Finally, we observed a strong correlation between the three metrics used for depressive-like behaviors – social interaction behaviors, sucrose preference and preference for female urine, in AD and A mice (Supplementary Figure 2c). Importantly, preference for sucrose and female urine did not correlate with the time spent in EPM open arms and in OFT center, the two metrics used in our study for anxiety-like behaviors (Supplementary Figure 2d).

Together, these new experimental results confirmed that social interaction phenotype as a reliable metric to reflect the depressive-like behaviors that CSDS induced in AD mice but not in A mice.

2. The correlation plots in Figure 2g-h are not terribly convincing, yet they are used to support the claim that VTA-BLA dopamine neuron firing rate differentiates anxiety and depression phenotypes. It is a relatively small number of animals, and the trend appears to be driven largely by one or two control mice. Is there any significant correlation between firing rate and open arm time if you only plot just the stressed animals?

We thank the reviewer for this comment. To address the reviewer's concern, we increased our sample size from n=13 mice to n=23 mice in Figure 2g-h.

Following the reviewer's insightful recommendation, we first confirmed that in control mice, VTA→BLA neuron firing rate does not correlate with social interaction behaviors but significantly correlates with the time spent in EPM open arms (Supplementary Figure 3h). Further, we did not observe in AD and A mice, the association between VTA→BLA neuron activity and the time spent in EPM open arms. These new experimental results are now described in the Results section (Line 172-175).

In conclusion, the contribution of VTA-BLA dopamine neuronal activity to anxiety- but not depressive-like, behaviors is corroborated by consistent evidence: alterations of firing activity (Figure 2f), intrinsic excitability (Figure 2j and k), and ion channel

properties (Figure 2l and m) in both AD and A mice after CSDS paradigm. These results are also confirmed by our fiber photometry measurements, showing that VTA-BLA neuronal dynamics are not associated with social interaction behaviors (Supplementary Figure 4d and e) but correlated with the time spent in EPM open arms before and after CSDS (Figure 3 and Supplementary Figure 5f and g). Finally, to fully support our hypothesis that VTA-BLA neurons control anxiety-like but not depressive-like behaviors, we performed additional optogenetic experiments (Supplementary Figure 8 and 10). We show that optogenetic VTA-BLA neuronal inhibition and stimulation bi-directionally induced and rescued anxiety-like behaviors respectively, the same manipulations failed to induced and rescued depressive-like behaviors, i.e. social interaction behavior and preference for sucrose and female urine.

3. It is generally accepted that the EPM is not a very “repeatable” behavioral assay, meaning that upon second exposure to the maze animals tend to dramatically reduce open arm exploration. It is therefore concerning that the authors perform EPM twice on the same animals before and after CSDS. This could present a confound for the fiber photometry results — changes in circuit activity interpreted as due to stress may simply be due to adaptation to the same maze. The stress naive control is important for addressing this confound, but from the data presented it is difficult to tell which data points in Figure 3e correspond to those in Figure 3i (i.e. how the control animals’ behavior and neural signals change upon second exposure to the EPM). Given the bizarre and surprising emergence of a large ‘center’-related photometry signal after CSDS, this is cause for concern that the change in signal may be related to repeated exposure to the EPM. The authors should carefully control for this potential confound.

We greatly appreciate the reviewer’s very insightful comments regarding potential confounds generated by repeated use of EPM. Because it has been reported that multiple exposures to EPM may impact the behavioral outcome of later EPM performance (i.e. modification of anxiolytic efficacy), we designed our fiber photometry measurements such that each EPM test was performed at least 18 days apart. As shown in Supplementary Figure 1l, stress-naïve mice did not significantly display different EPM performance at day 14 when compared to day -4. We strongly agree with the reviewer suggestion, therefore, the fiber photometry recordings of the control mice—initially shown in the Supplementary materials—are now presented in the main Results section as Figure 3k.

Lastly, in regard to the “surprising emergence of a large ‘center-related photometry signal”, we reproduced our data set with an additional cohort of 8 mice. Our analyses show that CTL mice have similar VTA-BLA neuronal dynamics during the first and second EPM tests, while CSDS alters the VTA-BLA neuronal dynamics with the emergence of EPM ‘center’-related photometry signal (Figure 3f, j and k).

4. In Figure 3i, for the post-CSDS correlation plots, it is difficult to see the A and AD points since they are squished together on the x-axis. Could the authors provide

additional plots of just the A and AD animals without controls so we can get a better sense of what these correlations look like in just the stressed animals?

We thank the reviewer for the recommendation concerning the post-CSDS correlation plots. We now provided additional plots in Supplementary Figure 5 separating CTL from AD & A mice to better visualize the data.

5. In Line 228 the authors state: “the anxiogenic effect... is context dependent and requires a pre-exposure to stressful events to induce the expression of anxiety-like behavior”. This is an intriguing result. Is it presumably because the stress is already inducing partial suppression of activity in these neurons? What would happen if you more completely inhibited VTA-BLA neurons in stress-naïve animals (e.g. greater than ~45% observed with this pulsed laser inhibition)?

We thank the reviewer for the opportunity to address this very interesting question, as we believe the related new results expand the scope of our study. We performed the suggested experiment and exposed stress-naïve mice to a more complete inhibition of the VTA-BLA neurons, i.e. 300 s, 582 nm laser on (Supplementary Figure 7d and e). This experiment identified that an anxiogenic effect was achieved when applying such strong NpHR-optogenetic modulation (1 Hz, 1 s pulse width) in stress-naïve mice. We greatly appreciate this reviewer’s recommendation. These new results suggest that the VTA-BLA hypoactivity does not require contextual association and pre-exposure to stress to induce to anxiety-like behaviors and may reveal a more general mechanism of VTA-BLA hypoactivity in encoding anxiety-like behaviors. Therefore, we have made changes accordingly in the revised manuscript (Line 276-280 and Line 401-404).

6. In Figure 4, does ChR2 stimulation differentially affect A and AD mice? Those groups are not differentiated in this figure.

We thank the reviewer for highlighting this important point. The related data set is now provided in Supplementary Figure 9c for AD mice and 9d for A mice. These results show that the ChR2 optogenetic stimulation of VTA-BLA neurons increase the time spent in EPM open arms in both AD and A mice.

Additionally, in a new cohort of mice in Supplementary Figure 10, we observed that VTA-BLA neuronal stimulation did not modulate social interaction/avoidance behaviors and preference for female urine but did increase the time spent in EPM open arms and OFT center in both AD-ChR2 mice (Supplementary figure 10b’) and A-ChR2 mice (Supplementary figure 10c’). These new experimental results further strengthen our hypothesis that the midbrain projection to the BLA contributes to controlling anxiety-like but not depressive-like behaviors.

7. Does sub-threshold social defeat stress not produce any behavioral effects on its own? In Figure 4c, it appears that the SubD eYFP group has similar if not even higher % time in open arms as the stress-naïve controls in previous experiments.

We thank the reviewer for the opportunity to address this important question. Compared to the other experiments, the mice exposed to sub-threshold social defeat stress were not required to be singled-house after the procedure and are returned in their home cage grouped-housed with their littermates, which may result in an overall lower level of anxiety when compared to the other experimental groups. This important information is now provided in the Results section (Line 266) in addition to the Methods section (Line 602) of the revised manuscript.

8. Figure 4 - locomotion controls should be prominently shown for all optogenetic experiments. Optogenetic manipulation of VTA projections is known to produce locomotor effects when strong enough stimulation parameters are used, which could be a confound for changes in open arm time. These results are provided in the supplemental data, and it looks like there may be a trend toward increased locomotion with ChR2 stimulation. What are the exact p-values for these locomotor plots in Supplementary Figure 6f and 6g?

As mentioned by the reviewer, these analyses were performed and are now documented in the Supplemental Figure 9f and 9g (previously 6f and 6g). Importantly, we did not observe statistically significant effects of the ChR2 stimulation on the locomotor activity across our study. Below we provided the exact statistical p-values for the reviewer's reference.

Figure	Group	Statistic value	P values
Sup. Figure 9e	CSDS eYFP/ChR2	t=0.2070; t=0.3042	P=0.9957; P=0.9867
Sup. Figure 9f	CSDS eYFP/ChR2	t=1.686; t=1.992	P=0.2854; P=0.1654
Sup. Figure 9g	CSDS eYFP/ChR2	t=1.101; t=0.862	P=0.6250; P=0.7788
Sup. Figure 10b	AD eYFP/ChR2	t=1.178; t=0.9787	P=0.2559; P=0.3423
Sup. Figure 10b'	AD eYFP/ChR2	t=1.158; t=1.135	P=0.2640; P=0.2733
Sup. Figure 10c	A eYFP/ChR2	t=2.198; t=1.554	P=0.0526; P=0.1513
Sup. Figure 10c'	A eYFP/ChR2	t=0.7220; t=0.8894	P=0.4868; P=0.3947
Sup. Figure 11b	CTL eYFP/ChR2	t=0.1060; t=0.2207	P=0.999; P=0.9949
Sup. Figure 11c	CTL eYFP/ChR2	t=0.1812; t=0.004086	P=0.9971; P>0.9999
Sup. Figure 11d	CTL eYFP/ChR2	t=0.1082; t=0.4785	P=0.9994; P=0.9538
Sup. Figure 11f	CTL eYFP/ChR2	t=0.2992; t=0.2921	P=0.7692; P=0.7745
Sup. Figure 11g	CTL eYFP/ChR2	t=0.5511; t=0.2726	P=0.5903; P=0.7889

9. Do these VTA-BLA projection neurons send collateral projections to any other region?

This is an interesting question, also raised by Reviewer #1. As mentioned by Reviewer #1, VTA-DA neurons projecting to the nucleus accumbens (NAc), the primary target of the VTA, sends extra-striatal axons to other brain areas. Also we assess if VTA neurons projecting to the BLA also send collaterals to the NAc, by performing viral mediated-circuit tracing (Supplementary Figure 3). We injected retrograding AAVs in the NAc allowing for somatic expression of Td-Tomato and retrograding AAVs in the BLA allowing for somatic expression of eGFP. We identified that only 2.7% amongst the 2100 VTA neurons labeled were positively labeled for both tdTomato and eGFP, i.e. NAc and BLA pathways. These anatomical results support that VTA-BLA projecting neurons don't send robust collaterals to the NAc.

Additionally, we and others (Ford et al., 2006; Lammel et al., 2008; Margolis et al., 2008) established distinct electrophysiological properties between VTA-NAc, VTA-BLA and VTA dopamine neurons projecting to the medial prefrontal cortex (mPFC). In particular, our current results and previous publications (Chaudhury et al., *Nature* 2013; Friedman et al., *Science* 2014) identify distinct physiological responses to CSDS upon the VTA neuron projection targets: VTA-BLA neurons are hypoactive in both AD and A mice, while VTA-NAc neurons are hyperactive in AD mice but not A mice. Moreover, VTA-BLA neurons of stress-naïve mice display I_h currents, while VTA-PFC neurons do not. In Fig. B presented above, we scheme our current and previous results supporting that VTA-BLA neurons primarily project to the BLA. Together, in agreement with previous reports (Ford et al., 2006; Lammel et al., 2008; Margolis et al., 2008), our results support the idea that there is a low probability for VTA-BLA neurons to have robust collateral projections to the NAc and the PFC.

Reviewer #3 (Remarks to the Author):

In this manuscript, the authors examine the intriguing question regarding separate circuits in CSDS mice that underlie depressive and anxiety phenotypes. They show that CSDS will induce depressive symptoms in 50% of the mice, but 100% show anxiety phenotypes. In this study, they use electrophysiology and fiber photometry/optogenetic methods to dissociate distinct dopamine pathways involved in these disorders, and show that the VTA DA system projecting to the amygdala is involved in the anxiety phenotype. This is a well-done study with potentially important results. Specific critique follows:

We thank the reviewer for the very positive and constructive comments.

1. The use of a single metric for depressive behavior based on social interaction is likely not adequate, as there are a number of conditions that can impact social interaction, particularly given that the inducing stimulus is based on a negative social interaction.

We thank reviewer for this constructive comment. Based on this suggestion, shared with Reviewer #2, we expanded the behavioral assessments used for depressive-like phenotypes following CSDS in both AD and A mice.

In a new cohort of mice presented in Supplementary Figure 2, we assessed sucrose preference as a reliable measurement for anhedonia, a major tenet of depressive-like behavior. In line with previous reports, we observed that AD mice (susceptible to depression), but not A mice (resilient to depressive outcomes), have a reduced sucrose preference when compared to control mice (Supplementary Figure 2b). Additionally, we performed a novel female urine-sniffing test (FUST) to measure reward-seeking behaviors. FUST measures the ability to elicit an approach behavior toward sexual related cue, i.e. urine (Malkesman et al., 2011). We observed that AD mice, when compared to stress-naïve control and A male mice, have a reduced interest in

pheromonal odors from the opposite sex. When we analyzed the association between social interaction ratio, sucrose preference and preference for female urine, we observed a strong correlation between the three metrics used for depressive-like behaviors in AD and A mice (Supplementary Figure 2c). Importantly, preference for sucrose and female urine does not correlate with the time spent in EPM open arms and in OFT center, the two metrics used in our study for anxiety-like behaviors (Supplementary Figure 2d).

Together, these new experimental results confirmed that CSDS induced depressive-like and anxiety-like behaviors in AD mice and singular anxiety-like behaviors in A mice. More importantly, these results further confirmed the use of the social interaction ratio as an accurate metric to reflect depressive-like behaviors induced by CSDS paradigm.

2. It is not valid to say that there was a “significant decrease” in VTA DA firing, when one is not recording from the same neurons before and after a manipulation; it could just as readily be two different populations of neurons. A more parsimonious account would be to say that the average firing rate of the neurons recorded is lower after CSDS.

We thank the reviewer for the constructive comment. We have now modified the Results section accordingly in our revised manuscript (Line 166).

3. Figure 2H – the correlation between firing rate and time in open arms is driven entirely by the control mice; using all mice for a single correlation is not valid. This is also the case in Figure 3, where the effect is driven entirely by the control mice, and not by a comparison of A and AD mice.

Acknowledging the reviewer comments, we modified our Result section to better reflect our observations and new analyses (Line 169-172): *We observed that VTA→BLA dopamine neuronal activity does not correlate with social interaction/avoidance behaviors ~~depressive-like behaviors~~ [...] but does correlate strongly with the time (%) spent in EPM open arms ~~anxiety-like behaviors~~.*

In our study, after defining that CSDS treated mice have lower firing activity when compared to control mice, we tested if the VTA-BLA neurons activity is associated with the overall time spent in EPM open arms or overall social interaction ratio. We included all individual behavioral responses in our correlation analyses. Additionally, following the reviewer’s insightful comments, we performed the correlation analyses separating control mice from CSDS treated mice (Supplementary Figure 3h-I). We first confirmed that in control mice, VTA→BLA neuron firing rate correlates with the time spent in EPM open arms (Supplementary Figure 3h). We then observed that CSDS alters the association between VTA→BLA neuron firing activity and the time spent in EPM open. These new results were described in the Results section (Line 172-175).

Regarding the correlation analyses shown in Figure 3, we first we would like to highlight that VTA→BLA neuron activities in freely behaving mice, correlate with the time (%)

spent in EPM open arms in mice before (Figure 3e) and after CSDS exposure (Figure 3i). To better visualize our results, we further analyzed the association between VTA→BLA neuron dynamic and time in EPM open arm separating control mice from CSDS exposed mice (Supplementary Figure 5f, g).

4. The effects of optogenetic inhibition/activation of the VTA-BLA on behavior in the open arm maze is interesting. However, this may be confounded by the fact that DA in the BLA is involved in learning aversive behavior. Is the result truly an anxiety effect or a learning effect?

We thank the reviewer for this extremely interesting question that we think can expand further the impact of our study. As described by the reviewer, VTA dopamine neurons projecting to the amygdala, as well as other brain areas such as the NAc, have been shown to contribute to negative and/or positive learning. Indeed dopamine plays a pivotal role encoding positive or negative stimuli and reinforcing properties that are critical in learning processes.

Aiming to address the reviewer's question, we first observed that while NpHR-mediated inhibition of VTA-BLA neurons reduces the time spent in EPM open arms, the anxiogenic effect was transient and ceased immediately when the 582nm laser was turned off in both sub-threshold socially defeated mice (Figure 4c, d) and stress-naïve mice (Supplementary Figure 6d and e). Importantly, we recapitulated these findings when performing an OFT test (Supplementary Figure 8b and c). We also tested if previous optogenetic modulation during the EPM test would impact the future EPM performance (Supplementary Figure 8d). We did not observe significant behavioral differences between eYFP-mice and NpHR-mice. Moreover, we performed an EPM test in CSDS-eYFP and CSDS-ChR2 mice a week after the last optogenetic stimulation (Supplementary Figure 10d and e) and again, we did not observe significant differences of the time spent in EPM open arms between CSDS-eYFP and CSDS-ChR2 mice. Our interpretation is that the transient impact of VTA-BLA neurons modulation on the mouse anxiety-like behaviors does not suggest a learning effect of such stimulations.

Lastly, in the context of exploratory behaviors, our time-locked fiber photometry measurements in control mice show that the VTA-BLA neurons activation upon transition to the open arms (Figure 3f) is conserved across EPM trials (Figure 3k). Hypothetically, the transient activation of VTA-BLA neurons may encode a drive towards exploratory behaviors rather than a learning signal.

References:

Beier, K. T. et al. Circuit Architecture of VTA Dopamine Neurons Revealed by Systematic Input-Output Mapping. *Cell* 162, 622–634 (2015).

Chaudhury, D. et al. Rapid regulation of depression-related behaviours by control of midbrain dopamine neurons. *Nature* 493, 532–536 (2013).

Fallon, J. H. Collateralization of monoamine neurons: Mesotelencephalic dopamine projections to caudate, septum, and frontal cortex. *J. Neurosci.* 1, 1361–1368 (1981).

Ford, C. P., Mark, G. P. & Williams, J. T. Properties and opioid inhibition of mesolimbic dopamine neurons vary according to target location. *J. Neurosci.* 26, 2788–2797 (2006).

Friedman, A. K. et al. Enhancing depression mechanisms in midbrain dopamine neurons achieves homeostatic resilience. *Science* (80). 344, 313–319 (2014).

Lammel, S. et al. Unique Properties of Mesoprefrontal Neurons within a Dual Mesocorticolimbic Dopamine System. *Neuron* 57, 760–773 (2008).

Malkesman, O. et al. The Female Urine Sniffing Test: A Novel Approach for Assessing Reward-Seeking Behavior in Rodents. *Biol Psychiatry* 67, 864–871 (2011).

Margolis, E. et al. Midbrain Dopamine Neurons: Projection Target Determines Action Potential Duration and Dopamine D2 Receptor Inhibition. *J. Neurosci.* 28, 8908–8913 (2008).

REVIEWERS' COMMENTS

Reviewer #1 (Remarks to the Author):

In this revised manuscript, Morel et al. addressed most of the concerns presented by the three reviewers. My main concerns were to question the method used to assign mice to different behavioral phenotypes, and to specify the cellular adaptations taking place in VTA-BLA neurons in the 3 experimental groups. The authors have conducted additional experiment demonstrating HCN channels as a potential molecular contributor for the VTA-BLA neuronal hypoactivity in AD and A mice. Moreover, authors have performed additional and convincing experiments to address the possibility that the VTA-BLA neurons also have a collateral to the NAC. However, it remains difficult to apprehend the proportion of dopaminergic-VTA-BLA and non-dopaminergic VTA-BLA neurons when the viral retrograde strategy is used in a C57BL6 mice. Altogether, the authors appear to have put together a thoroughly revised manuscript that has significantly improved their paper.

Reviewer #3 (Remarks to the Author):

The author has addressed all of my critiques

** See Nature Research's author and referees' website at www.nature.com/authors for information about policies, services and author benefits

Responses to the reviewers' comments for the revision of NCOMMS-20-44686B

General remark

We would like to begin by thanking the reviewers for their stimulating comments. They have led us to improve and clarify our manuscript greatly. Thanks to these changes, we provide stronger evidence and a more informed discussion of our main findings. We are also pleased that the reviewers feel their critics have been addressed; they led us to include important data and analyses, in particular concerning the characterization of the stress-induced behavioral phenotypes and the associated neural activity and dynamics, and so helped substantially improve the manuscript.

Reviewer #1 (Remarks to the Author):

In this revised manuscript, Morel et al. addressed most of the concerns presented by the three reviewers. My main concerns were to question the method used to assign mice to different behavioral phenotypes, and to specify the cellular adaptations taking place in VTA-BLA neurons in the 3 experimental groups. The authors have conducted additional experiment demonstrating HCN channels as a potential molecular contributor for the VTA-BLA neuronal hypoactivity in AD and A mice. Moreover, authors have performed additional and convincing experiments to address the possibility that the VTA-BLA neurons also have a collateral to the NAC. However, it remains difficult to apprehend the proportion of dopaminergic-VTA-BLA and non-dopaminergic VTA-BLA neurons when the viral retrograde strategy is used in a C57BL6 mice. Altogether, the authors appear to have put together a thoroughly revised manuscript that has significantly improved their paper.

We thank the reviewer's very positive comments on our revision. We also sincerely thank the reviewer for the appreciation of our newly added HCN channel mechanism for the cellular adaptations observed in VTA-BLA neurons in the control, A/D and A groups. As to the proportion of DA and non-DA VTA-BLA neurons, we performed several projection-specific quantifications of our viral strategy used to monitor and manipulate the VTA-BLA DA neuronal activity. We demonstrated that in C57BL/6J mice 82-87% of the labeled neurons expressed the eGFP (or eYFP) reporter in TH⁺ neurons, suggesting that the large proportion of neurons monitored and manipulated when using C57BL/6J mice were dopaminergic VTA-BLA neurons. However, we are very grateful for the reviewer's insightful comments, as future new experiments answering this question will help us to refine the detailed DA and non-DA subcircuit components and their specific functions of VTA-BLA neurons, which would provide a new opportunity for us to develop another story.

Reviewer #3 (Remarks to the Author):

The author has addressed all of my critiques

We sincerely thank the reviewer for the original insightful and constructive comments, which have helped us tremendously to refine our experimental design and expand the scope of our study.